# MoSE: Mixture of Slimmable Experts for Efficient and Adaptive Language Models

**Nurbek Tastan** [1]  **Stefanos Laskaridis** [†2]  **Karthik Nandakumar** [1 3]  **Samuel Horváth** [1]

## Abstract

Mixture-of-Experts (MoE) models scale large language models efficiently by sparsely activating experts, but once an expert is selected, it is executed fully. Hence, the trade-off between accuracy and computation in an MoE model typically exhibits large discontinuities. We propose Mixture of Slimmable Experts (MoSE), an MoE architecture in which each expert has a nested, slimmable structure that can be executed at variable widths. This enables conditional computation not only over which experts are activated but also over how much of each expert is utilized. Consequently, a single pretrained MoSE model can support a more continuous spectrum of accuracy-compute trade-offs at inference time. We present a simple and stable training recipe for slimmable experts under sparse routing, combining multi-width training with standard MoE objectives. During inference, we explore strategies for runtime width determination, including a lightweight test-time training mechanism that learns how to map router confidence/probabilities to expert widths under a fixed budget. Experiments on GPT-style models, various routing regimes, zero-shot downstream reasoning benchmarks, and continual pre-training adaptation of DeepSeek model show that MoSE matches or improves standard MoE at full width and consistently shifts the compute-quality frontier toward lower inference FLOPs. The code can be found at: https://github.com/tnurbek/mose.

## 1. Introduction

Large language models (LLMs) continue to benefit from scale, but the associated training and inference costs (compute, energy, and latency) increasingly constrain their deployment and accessibility (Radford et al., 2019; Brown et al., 2020; Grattafiori et al., 2024). Mixture-of-Experts (MoE) architectures address this challenge by sparsely activating experts, allowing parameter counts to grow without proportional increases in per-token computation. This conditional computation principle dates back to sparsely-gated MoE layers (Jordan & Jacobs, 1994; Shazeer et al., 2017) and has been refined in large-scale systems such as switch transformers and subsequent MoE LLMs (Fedus et al., 2022; DeepSeek-AI et al., 2024a; Jiang et al., 2024; DeepSeek-AI et al., 2024b; Agarwal et al., 2025; Yang et al., 2025). As a result, MoE models have become a dominant paradigm for scaling LLMs efficiently (Du et al., 2022; Jiang et al., 2024; Agarwal et al., 2025).

However, despite sparse expert selection, once an expert is activated, it is almost always executed at full capacity. This design choice leaves a substantial dimension of conditional computation unexplored: the ability to adapt the amount of computation allocated within each activated expert. In practice, different tokens, contexts, and experts vary widely in their computational needs, suggesting that expert capacity itself should be elastic rather than fixed.

We introduce **Mixture of Slimmable Experts (MoSE)**, an MoE architecture that equips each expert with slimmable widths via nested subnetworks. MoSE generalizes conditional computation in MoE models by decoupling expert **selection** from expert **capacity**: the router determines which experts are active, while execution width controls how much of each expert is used. This enables a single pretrained model to flexibly trade off accuracy and compute at inference time, without retraining or modifying expert parameters. For example, one can:

(i) run all activated experts at a uniform width to obtain a smooth quality-compute curve, or

(ii) adapt widths per token based on router outputs.

Beyond uniform width control, MoSE enables adaptive width allocation conditioned on router confidence. We further propose a lightweight **test-time training** procedure that learns a low-dimensional mapping from router outputs to expert widths under a fixed compute budget, while keep-

---

[†]Work done independently of Amazon. [1]Mohamed bin Zayed University of Artificial Intelligence (MBZUAI), UAE [2]Amazon Science, UK [3]Michigan State University (MSU), USA. Correspondence to: Nurbek Tastan <nurbek.tastan@mbzuai.ac.ae>.

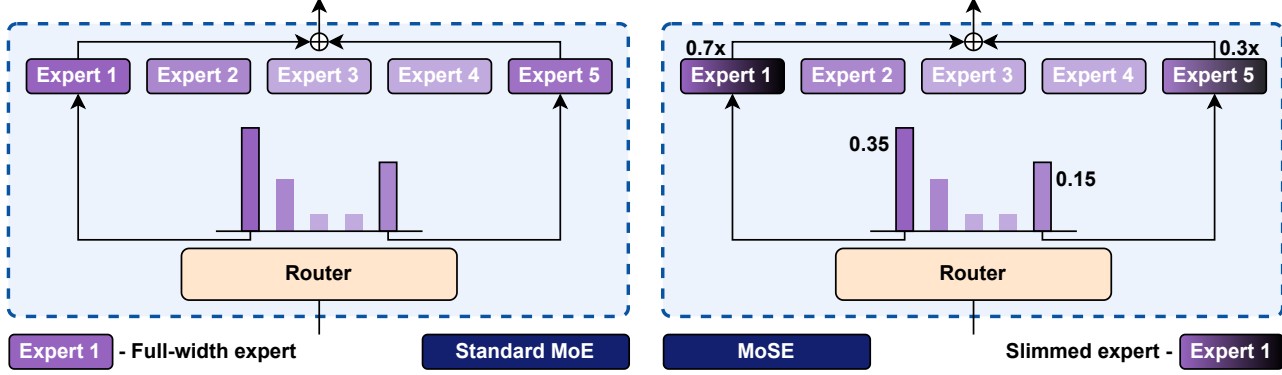

*Figure 1.* Comparison between standard MoE and MoSE. In the standard case, the router selects a fixed number of full-width experts and activates all their parameters. In the proposed method, the router not only selects multiple experts but also adjusts their widths, allowing more experts to contribute under the same parameter budget. This increases expert diversity without increasing total compute cost, potentially improving model accuracy at the same efficiency.

ing all model parameters frozen. This allows MoSE to automatically allocate computation where it is most beneficial at inference time, yielding consistent improvements in compute-quality trade-offs with negligible adaptation cost.

MoSE is especially natural for MoE models because routing probabilities already provide a token-specific signal of expert importance. Experts with lower routing probabilities contribute less to the final output and can therefore be executed at reduced widths with a limited effect on quality. At the same time, expert FFNs dominate MoE-layer computation due to their large hidden dimension expansion. Width adaptation, therefore, targets the main compute bottleneck while leveraging information that is already present in the routing distribution. We make the following contributions:

- We propose **MoSE**, an MoE architecture with slimmable experts that enables conditional computation over both expert selection and expert capacity.
- We present a simple and stable training recipe for slimmable experts under sparse routing, requiring no changes to the standard MoE pipeline.
- We introduce inference-time width allocation strategies, including a lightweight **test-time training** method that learns compute-aware width assignment under fixed budgets.
- Through extensive experiments on GPT-style and modern pretrained MoE models, including fine-grained routing regimes, zero-shot downstream reasoning tasks, and continual-pretraining adaptation, we show that MoSE consistently improves the compute-quality frontier at comparable or lower inference FLOPs.

## 2. Methodology

### 2.1. Mixture-of-Experts

We consider a decoder-only transformer in which the feed-forward network (FFN) in each block is replaced by an MoE

layer. The MoE layer consists of $n$ experts $(\mathcal{E}_1, \mathcal{E}_2, \ldots, \mathcal{E}_n)$ and a router/gating network $\mathcal{G}$. The experts are neural networks, each with their own parameters. For a token representation (input) $\boldsymbol{x} \in \mathbb{R}^d$, a gating network $\mathcal{G}$ produces scores $\mathcal{G}(\boldsymbol{x})$ over $n$ experts and selects a sparse subset of experts, thereby saving computations through the sparsity of $\mathcal{G}(\boldsymbol{x})$ (typically using top-$k$). Let us denote $\mathcal{E}_i(\boldsymbol{x})$ as the output of the $i$-th expert network for the given input $\boldsymbol{x}$. The output $\boldsymbol{y}$ of the MoE module can be written as follows:

$$\boldsymbol{y} = \sum_{i=1}^{n} \mathcal{G}(\boldsymbol{x})_i \mathcal{E}_i(\boldsymbol{x}). \quad (1)$$

We save computation whenever $\mathcal{G}(\boldsymbol{x})_i = 0$, since we do not need to compute $\mathcal{E}_i(\boldsymbol{x})$.

### 2.2. MoSE: Mixture of Slimmable Experts

MoSE extends the standard MoE formulation by equipping each expert with **slimmable** (ordered) widths. Instead of executing a selected expert at full capacity, MoSE allows for controlling how much of each expert is used by adjusting its internal width. This introduces an additional axis of conditional computation: beyond selecting which experts are active, the model can adapt the **capacity** of each active expert.

Each expert $\mathcal{E}_i$ in MoSE is implemented as a transformer FFN with an expansion ratio of $4$ (most language models follow this setting (Radford et al., 2019; Fedus et al., 2022; Touvron et al., 2023)), consisting of two fully connected layers. For $\boldsymbol{x} \in \mathbb{R}^d$, a full-width expert computes:

$$\mathcal{E}_i(\boldsymbol{x}) = \phi\left(\boldsymbol{x} W_{\text{up}}^{(i)} + b_{\text{up}}^{(i)}\right) W_{\text{down}}^{(i)} + b_{\text{down}}^{(i)}, \quad (2)$$

where $W_{\text{up}}^{(i)} \in \mathbb{R}^{d \times 4d}$ and $W_{\text{down}}^{(i)} \in \mathbb{R}^{4d \times d}$, $\phi(\cdot)$ denotes a nonlinear activation function (e.g., GELU (Hendrycks & Gimpel, 2016)). The intermediate hidden dimension dominates the computational cost of the expert, making it the natural target for slimmability.

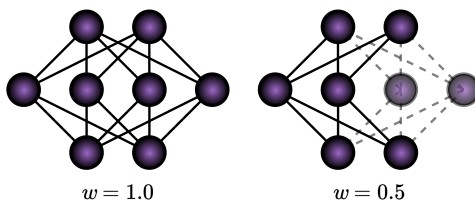

$w = 1.0 \qquad w = 0.5$

*Figure 2.* Slimmable expert in MoSE. Example with $w = 0.5$, where only half of the intermediate units of the expert FFN are activated by slicing the hidden dimension.

To enable slimmability, we define a discrete set of width multipliers (this can also be a continuous set):

$$\mathcal{A} = \{w_1, \ldots, w_m\}, \quad 0 < w_1 < \cdots < w_r = 1.0, \quad (3)$$

and let $m(w) = \lceil w \cdot 4d \rceil$ denote the active hidden size. The width-$w$ version of expert $\mathcal{E}_i$, denoted as $\mathcal{E}_i^w$, is obtained by slicing the intermediate dimension:

$$\begin{aligned}
W_{\text{up}}^{(i)}(w) &= W_{\text{up}}^{(i)}[:, : m(w)], \\
W_{\text{down}}^{(i)}(w) &= W_{\text{down}}^{(i)}[: m(w), :].
\end{aligned} \quad (4)$$

Figure 2 illustrates this mechanism for $w = 0.5$. All widths share parameters through this nested structure.

**MoSE forward computation.** Given $x$, the gating network $\mathcal{G}$ selects a sparse set of experts as in standard MoE (top-$k$). For each selected expert $\mathcal{E}_i$, MoSE additionally assigns a width $w \in \mathcal{A}$. The MoSE output is then

$$y = \sum_{i=1}^{n} \mathcal{G}(x)_i \mathcal{E}_i^{w(x)}(x), \quad (5)$$

where computation is skipped whenever $\mathcal{G}(x)_i = 0$, and the cost of each active expert scales with $m(w(x))$. When all $w(x) = 1.0$, MoSE reduces to standard MoE.

### 2.3. MoSE Pre-Training

Pre-training must ensure that experts perform well across multiple widths while preserving stable MoE routing. We optimize the standard language modeling objective, augmented with MoE-specific regularization and stochastic multi-width training.

To train a single MoSE model that is usable at many widths while keeping training overhead minimal, we sample two widths per iteration: (i) the full width $w_{\max}$ and (ii) one random width $w \sim \text{Uniform}(w_{\min}, w_{\max})$. Concretely, for each mini-batch, we run the model twice and backpropagate through the sum (or average) of the two losses. This strategy is inspired by random width training (Tastan et al., 2025; Horváth et al., 2021) in slimmable networks and provides a simple pre-training mechanism with low overhead compared to other sampling strategies that involve more

than two widths at a time. Let $\mathcal{L}_{\text{LM}}(w)$ denote the standard autoregressive next-token loss evaluated when all MoSE experts run at width $w$. Our primary objective is

$$\begin{aligned}
\mathcal{L}_{\text{LM}}^{\text{MoSE}} &= \frac{1}{2}\big(\mathcal{L}_{\text{LM}}(w_{\max}) + \mathcal{L}_{\text{LM}}(w)\big), \\
w &\sim \text{Uniform}(w_{\min}, w_{\max}).
\end{aligned} \quad (6)$$

As in standard sparsely-gated MoE training (Shazeer et al., 2017), we incorporate auxiliary losses to stabilize routing and prevent expert collapse. Specifically, the auxiliary objective includes a load balancing loss, which encourages an even distribution of tokens across experts within a batch, and a router z-loss (Zoph et al., 2022), which penalizes large router logits and improves numerical stability.

Notably, although the training uses uniform widths, results show that the learned representations generalize well to non-uniform, router-conditioned allocations at inference time.

**Practical regimes.** Each MoSE training step evaluates one full-width configuration and one sampled width configuration, so the per-step overhead depends on the sampled width and is lower than that of two full-width passes. We consider two practical usage regimes: (i) **full pretraining**, which yields the strongest compute-quality frontier, and (ii) **continual pretraining (CPT)** adaptation on top of an existing pretrained MoE checkpoint, which provides a cheaper path to enable slimmability post hoc.

### 2.4. MoSE Inference

After pre-training, MoSE supports multiple inference-time execution modes that differ in how widths are assigned to activated experts. Throughout, let $\mathcal{S}(x)$ denote the sparse set of selected experts (i.e., the experts with non-zero routing weight, $\mathcal{G}(x) \neq 0$). We assign widths only to the experts in this sparse set $\mathcal{S}(x)$.

**(1) Uniform-width execution.** The simplest mode uses a single global width $w \in [w_{\min}, w_{\max}]$ (or $w \in \mathcal{A}$ if discretized) for all activated experts ($w(x) = w$ in Equation 5). This provides a stable and easily controlled compute-quality trade-off and matches the width control used during pre-training.

**(2) Router-conditioned widths (normalized probability mode).** We next assign widths using the router probabilities. Let $p_i(x)$ denote router probabilities normalized over the selected experts:

$$p_i(x) = \frac{\mathcal{G}(x)_i}{\sum_{j \in \mathcal{S}(x)} \mathcal{G}(x)_j}, \quad i \in \mathcal{S}(x). \quad (7)$$

Given a sharpness parameter $\gamma > 0$, we define un-normalized width scores:

$$s_i(x; \gamma) = p_i(x)^\gamma, \quad i \in \mathcal{S}(x), \quad (8)$$

and normalize them into allocation weights:

$$q_i(\boldsymbol{x}; \gamma) = \frac{s_i(\boldsymbol{x}; \gamma)}{\sum_{j \in \mathcal{S}(\boldsymbol{x})} s_j(\boldsymbol{x}; \gamma)}. \qquad (9)$$

Intuitively, $\gamma$ controls how concentrated the width allocation is: $\gamma \to 0$ approaches uniform allocation across the selected experts, while larger $\gamma$ increasingly concentrates width on higher-probability experts. In this normalized-probability mode, we fix $\gamma = 1.0$, yielding a direct proportional mapping from router probabilities to expert width. In contrast, the test-time training mode learns $\gamma$ from data, allowing the probability-to-width mapping to adapt to the compute budget.

**Budgeted width assignment.** Given a per-token width budget $\Gamma \in (0, |\mathcal{S}(\boldsymbol{x})|)$ measured in "full-expert widths" (e.g., $\Gamma = |\mathcal{S}(\boldsymbol{x})|$ matches standard MoE with full width experts), we assign widths:

$$\tilde{w}_i(\boldsymbol{x}; \gamma) = \Gamma \cdot q_i(\boldsymbol{x}; \gamma), \quad i \in \mathcal{S}(\boldsymbol{x}). \qquad (10)$$

We then map $\tilde{w}_i$ into a valid execution width by enforcing bounds and (optionally) discretizing (Equation 3):

$$w_i(\boldsymbol{x}) = \mathrm{Proj}_{[w_{\min}, w_{\max}]}(\tilde{w}_i(\boldsymbol{x}; \gamma)), \qquad (11)$$

where $\mathrm{Proj}$ denotes a simple projection/clipping.

Intuitively, $\Gamma$ controls how much total compute is available for a token, while $\gamma$ controls how that compute is distributed across the selected experts.

**(3) Test-time training for width identification.** Finally, we propose a lightweight test-time training (TTT) mechanism that learns how to map router probabilities to widths by optimizing the single scalar $\gamma$ (shared across layers) or a small set $\{\gamma_\ell\}$ (layer-specific), while keeping all model and expert weights fixed. Concretely, for a short adaptation stream $\mathcal{D}_{\mathrm{calib}}$, we minimize the language modeling loss under the router-conditioned width policy:

$$\gamma^\star = \arg\min_{\gamma > 0} \mathbb{E}_{\boldsymbol{x} \in \mathcal{D}_{\mathrm{calib}}} [\mathcal{L}_{\mathrm{LM}}(\gamma; \Gamma)] \qquad (12)$$

where $\mathcal{L}_{\mathrm{LM}}(\gamma; \Gamma)$ is computed by running MoSE with widths set via Equations 7-10 under a chosen budget $\Gamma$. Since $\gamma$ is extremely low-dimensional, this adaptation is lightweight and is performed without modifying the pretrained model parameters. In deployment, we then fix $\gamma = \gamma^\star$ and use it to assign widths for subsequent inputs.

## 3. Experiments

### 3.1. Experimental Setup

All experiments are conducted using a sparsely-gated mixture-of-experts (MoE) training framework (Shazeer et al., 2017). Models are pre-trained on the OpenWebText

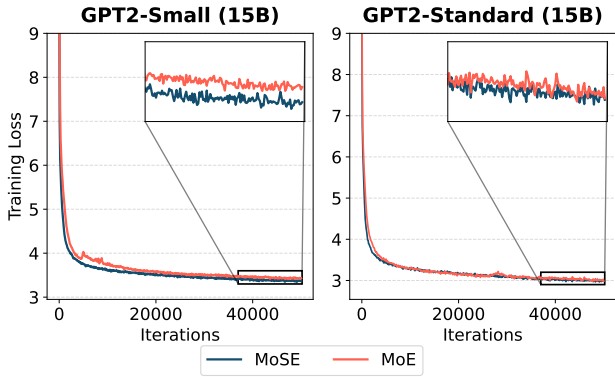

*Figure 3.* Pre-training dynamics of MoE and MoSE on OpenWeb-Text dataset using GPT2-Small model.

corpus (Gokaslan et al., 2019) using top-$k$ routing, and all MoSE models are trained using uniform-width execution.

We evaluate MoSE across a range of model scales and routing granularities, spanning GPT2-Small (55M), GPT2-Standard (322M), GPT2-Medium (1B), and GPT2-Large (2.5B) (Radford et al., 2019), together with finer-grained expert settings (E64A8 and E128A8) and continual-pretraining adaptation on top of pretrained MoE checkpoints, including DeepSeek-V2-Lite (16B) (DeepSeek-AI et al., 2024a). Unless otherwise specified, the main experiments use routing configurations E8A2, E8A4, and E16A4 with training budgets ranging from 3B to 15B tokens. Exact architectural parameters, routing hyperparameters, and per-experiment training schedules are provided in Appendix A.

After pre-training, models are evaluated under several inference-time execution modes (detailed in Section 2.4), including (1) uniform-width execution, (2) router-conditioned widths with fixed $\gamma$, and (3) test-time training for widths, where the sharpness parameter $\gamma$ (shared or layer-wise) is learned using a short calibration stream while keeping all pretrained weights fixed. Inference-time compute is reported in MFLOPs per token.

We report language modeling perplexity on OpenWebText (Gokaslan et al., 2019) and WikiText-103 (Merity et al., 2017), as well as zero-shot performance on LAMBADA (accuracy and perplexity) (Paperno et al., 2016), the Winograd Schema Challenge (WSC) (Levesque et al., 2012), HellaSwag (Zellers et al., 2019), PIQA (Bisk et al., 2020), and Social IQA (Sap et al., 2019). All downstream evaluations are performed without task-specific fine-tuning. Further experimental details are deferred to Appendix A.

### 3.2. MoSE Pre-Training Results

In this section, we study the pre-training behavior of MoSE on the OpenWebText dataset, focusing on optimization stability and convergence relative to the MoE baseline.

*Table 1.* Pre-training and zero-shot evaluation results of MoE and MoSE across model sizes and training budgets. Results are reported on OpenWebText and WikiText-103 for language modeling perplexity, and on LAMBADA and Winograd Schema Challenge (WSC) for zero-shot evaluation. * denotes lower FLOPs at comparable or better performance (Section 3.3). The best results are highlighted in bold.

| MODEL | TOKENS | METHOD | OPENWEBTEXT [PPL] | WIKITEXT-103 [PPL] | LAMBADA [ACC] | LAMBADA [PPL] | WSC [ACC] |
|---|---|---|---|---|---|---|---|
| GPT2-SMALL (55M) | 3B | MoE | 41.82 | 160.13 | 0.152 | 123.527 | 0.4982 |
| | | MoSE ($w=1.0$) | 39.02 | 154.85 | 0.145 | 113.031 | 0.5092 |
| | | MoSE* (TTT) | **38.48** | **153.24** | **0.164** | **110.129** | **0.5201** |
| | 15B | MoE | 31.03 | 120.48 | 0.187 | 65.413 | 0.5165 |
| | | MoSE ($w=1.0$) | 31.26 | 121.50 | 0.203 | **50.437** | 0.5421 |
| | | MoSE* (TTT) | **30.81** | **120.20** | **0.214** | 51.085 | **0.5458** |
| GPT2-STANDARD (322M) | 3B | MoE | 27.75 | **89.22** | 0.206 | 39.808 | **0.5201** |
| | | MoSE ($w=1.0$) | 27.66 | 89.56 | 0.217 | 39.075 | 0.5128 |
| | | MoSE* (TTT) | **27.32** | 89.41 | **0.224** | **39.060** | 0.5165 |
| | 15B | MoE | 20.81 | 77.22 | 0.294 | 18.280 | 0.5311 |
| | | MoSE ($w=1.0$) | 20.65 | **76.93** | 0.336 | 16.516 | 0.5495 |
| | | MoSE* (TTT) | **20.38** | 76.99 | **0.338** | **16.430** | **0.5531** |

**Training dynamics and stability.** Figure 3 shows the evolution of training loss as a function of iterations for MoE and MoSE under 15B token budget. Across both settings, MoSE closely tracks the convergence trajectory of MoE throughout training. This behavior indicates that introducing width-adaptive execution does not hinder optimization or destabilize pre-training. These results show that MoSE can be trained end-to-end within standard MoE pre-training pipelines without additional modifications.

**Last-iterate pre-training performance.** Table 1 summarizes the outcomes of MoE and MoSE across model sizes and training token budgets, evaluated on OpenWebText (test set) as well as multiple downstream benchmarks. For MoSE, we report two variants: (i) uniform-width execution ($w = 1.0$), which assigns full width to all activated experts, and (ii) MoSE with test-time training for width identification (with a maximum budget $\Gamma = |\mathcal{S}(x)|$). For the latter, we report the layer-wise $\gamma$ variant throughout.

Across all settings, MoSE with uniform execution already matches or improves upon the MoE baseline, indicating that the architectural flexibility introduced by MoSE does not degrade representational capacity. More notably, when coupled with test-time training, MoSE consistently achieves the best performance (despite using lower FLOPs; discussed in later sections), yielding lower perplexity on OpenWebText and WikiText-103, as well as improved zero-shot performance on LAMBADA and WSC. These gains hold across both 3B and 15B token regimes and scale with model size, showing that the learned width identification mode complements pre-training rather than interfering with it.

### 3.3. MoSE Inference Results

In this section, we evaluate MoSE in the inference-time regime, where expert routing is fixed by a pre-trained router and the model dynamically assigns execution widths to the activated experts. Unless otherwise specified, all experiments report compute-quality trade-offs measured in MFLOPs per token versus validation perplexity, and Pareto frontiers are constructed by varying the width budget while keeping model parameters fixed. We compare MoSE (TTT) against uniform-width execution and vanilla MoE.

**Scaling across model sizes (E8A2).** Figure 4 compares MoSE across GPT2-Small, GPT2-Standard, and GPT2-Medium models under a fixed routing setting (E8A2; $n = 8, k = 2$). Across all model sizes, MoSE (TTT) consistently shifts the Pareto frontier downward relative to uniform-width execution, achieving lower perplexity at comparable compute. This demonstrates that the proposed width identification mechanism scales effectively with model size and does not rely on model-specific tuning. Notably, the relative improvement remains stable as the base model grows, indicating that test-time training learns a compute-aware width allocation strategy that works across scales.

**Scaling pre-training tokens at fixed routing.** Figure 5 studies the effect of increasing the amount of pre-training data while keeping the routing configuration fixed (E8A2). We increase the number of pre-training tokens from 3B to 15B for GPT2-Small and GPT2-Standard and evaluate MoSE using the same inference-time width allocation procedure. As shown, increasing the amount of pre-training data consistently shifts the Pareto frontier downward, improving perplexity at all compute budgets. Importantly, the advantage of test-time training over alternative inference modes remains stable under increased data scale, suggesting that the learned width identification captures structural properties of the routing-compute trade-off rather than overfitting to a particular data regime.

**Effect of routing setting and expert count.** Figure 6 analyzes the effect of the routing setting on MoSE using GPT2-Small, comparing E8A2, E8A4, and E16A4 configurations.

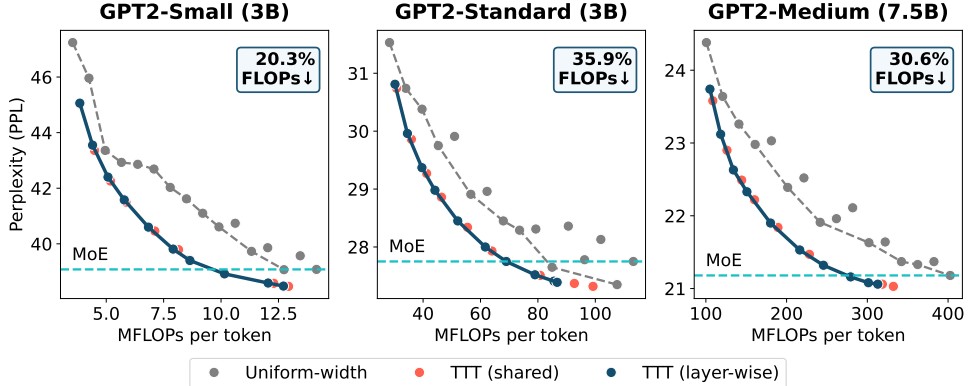

Figure 4. Compute-quality trade-offs across GPT2-Small, Standard, and Medium models under **E8A2** setting. MoSE with test-time training learns compute-aware width identification that shift the Pareto frontier, achieving lower perplexity than **uniform-width** mode at comparable MFLOPs per token.

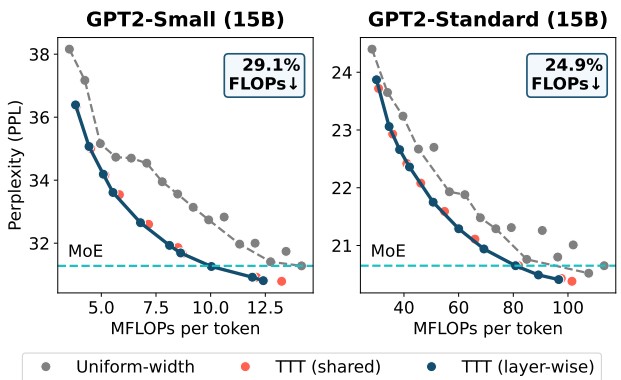 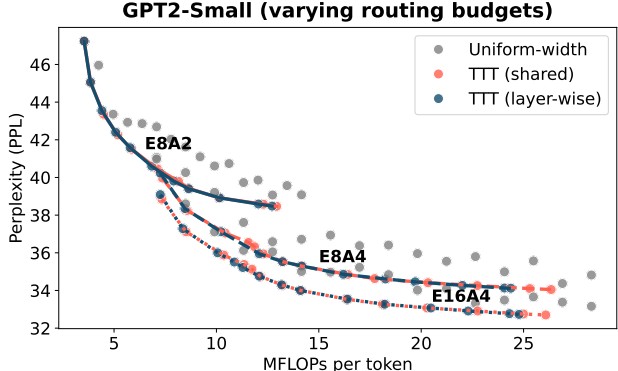

Figure 5. Scaling pre-training tokens at E8A2 setting. We scale the number of pre-training tokens from 3B (Figure 4) to 15B for GPT2-Small and GPT2-Standard, while keeping the same routing setup and compute budget. Our test-time training for MoSE width identification continues to dominate across the Pareto frontier, and increasing the amount of pre-training data consistently shifts the quality-compute trade-off downward. The relative advantage of test-time training (with both layer-wise and shared parameters) remains stable under increased data scale.

Figure 6. Effect of routing setting on MoSE with test-time width identification (GPT2-Small). We evaluate MoSE under different routing settings (E8A2, E8A4, E16A4) using GPT2-Small model. Points denote individual configurations, while solid curves trace the Pareto frontiers of respective modes. Increasing the budget and the expert count consistently shifts the Pareto frontier downward, achieving lower perplexity at comparable FLOP consumption.

Increasing both the width budget and the number of activated experts consistently improves the compute-quality trade-off, shifting the Pareto frontier downward. Across all settings, MoSE with test-time width identification maintains strong performance, indicating that the method adapts naturally to different routing granularities. This result highlights that width identification complements routing capacity: larger budgets provide more flexibility, which MoSE effectively exploits at inference time.

**Inference-time routing transfer.** Figure 7 examines the robustness of MoSE to inference-time routing changes when using a fixed pretrained checkpoint. We train a single MoSE model under the E16A4 routing configuration and evaluate it at inference time with fewer activated experts (E16A2, E16A3, and E16A4) without any retraining. The first three panels report Pareto frontiers for each evaluation setting independently, while the rightmost panel overlays the resulting frontiers to enable direct comparison.

As shown in Figure 7, MoSE with test-time training degrades gracefully as the number of active experts is reduced, maintaining strong perplexity-compute trade-offs even under more restrictive routing. Notably, E16A3 achieves similar performance to E16A4 at lower compute, indicating that activating more experts can be redundant once a moderate routing capacity is reached. This suggests diminishing returns from increasing the number of activated experts beyond this point for this experiment.

To identify the most suitable $k$, evaluating routing settings under full-width execution is sufficient, as the resulting Pareto frontier closely follows the best configuration.

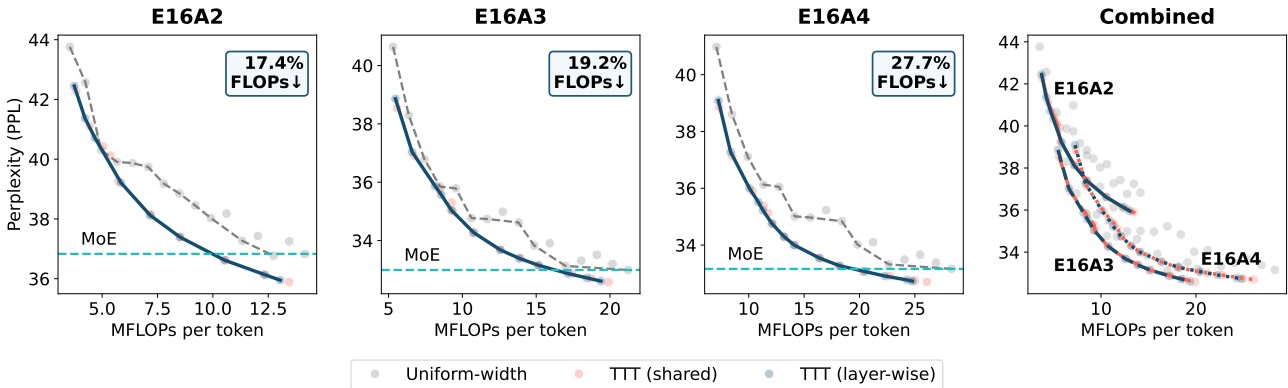

*Figure 7.* Inference-time routing transfer with a fixed checkpoint (GPT2-Small under n16k4 setting). This experiment evaluates a single MoSE checkpoint trained with a setting **E16A4** under different numbers of activated experts. The first three panels report results for each evaluation setting independently, while the fourth plot overlays Pareto frontiers of MoSE (TTT). Notably, E16A3 achieves essentially the same performance as E16A4 at reduced compute, indicating that E16A3 provides a more compute-efficient operating point for evaluation.

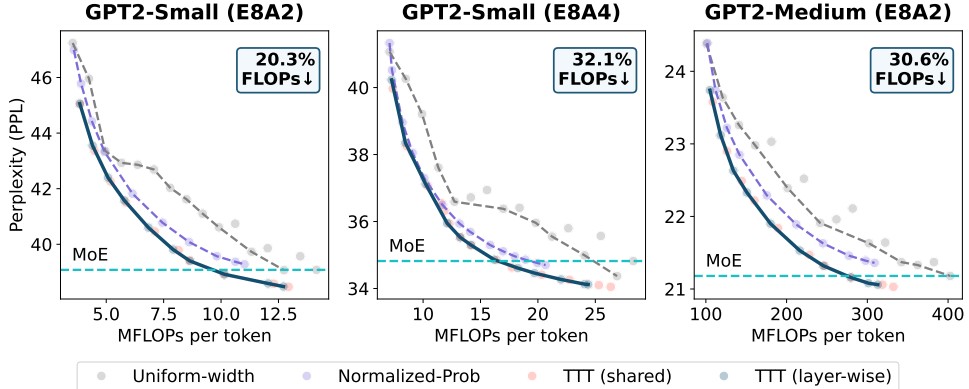

*Figure 8.* Ablation of MoSE inference modes. This figure compares the three inference-time execution modes supported by MoSE. Results are shown across different expert settings and model scales. Points denote individual configurations, while curves trace Pareto frontiers. The normalized-probability mode improves over uniform widths but remains consistently inferior to the learned TTT variant, indicating that performance gains are not solely due to probability-based allocation but require learning the probability-width mapping.

**Ablation study of MoSE inference modes.** Figure 8 isolates the effect of our proposed inference-time width assignment strategy in MoSE by comparing uniform-width identification, router-conditioned widths based on normalized probabilities (constant $\gamma = 1.0$), and test-time training (TTT) for width identification. Across model scales and expert settings, the normalized-probability baseline consistently improves over uniform widths, confirming that router confidence contains useful information for allocating compute. However, its Pareto frontier remains uniformly dominated by the learned TTT variants. Learning the sharpness parameter $\gamma$ at test time, either as a single shared value or in a layer-wise form, yields systematically better perplexity-compute trade-offs, demonstrating that effective width allocation requires adapting the probability-to-width mapping (given the budget) rather than relying on fixed normalization alone. The gap is most pronounced at lower compute budgets, where the accurate concentration of width on a subset of experts is critical, while shared $\gamma$ already captures much

of the gain, and the layer-wise variant provides additional flexibility and is more expressive.

**Transferability of $\gamma$.** Figure 9 reports zero-shot transfer performance on LAMBADA, using sharpness parameters $\gamma$ calibrated on OpenWebText and reused without modification. We also report our results in terms of accuracy and perplexity. As shown in the left panel, both learned variants consistently outperform the uniform-width MoSE across compute budgets, with MoSE (layerwise $\gamma$) achieving the highest accuracy at moderate MFLOPs. The accuracy curves exhibit mild non-monotonicity and local fluctuations, which are expected given that accuracy is a discrete metric and is sensitive to sampling noise and prediction discretization (Schaeffer et al., 2023).

In contrast, the right panel (perplexity) provides a smoother and more stable signal. Both test-time training modes yield consistently lower perplexity than the non-adapted variant across the full compute range, with layerwise $\gamma$ forming the

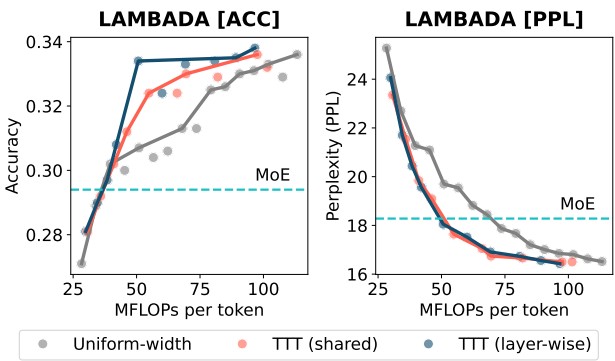

*Figure 9.* Transferability of the learned sharpness parameter $\gamma$ on LAMBADA (GPT2-Standard, 15B tokens).

dominant Pareto frontier. The steady downward trend in perplexity confirms that the transferred $\gamma$ values preserve their intended effect on width allocation, even under dataset shift. Together, these results indicate that the learned sharpness parameter generalizes effectively across datasets, improving zero-shot performance without test-time training, while perplexity serves as the more reliable indicator of this transfer behavior.

### 3.4. Extended Evaluation: Scale, Adaptation

We next extend the empirical scope of MoSE along three axes that are important for practical deployment: (i) larger backbones and finer-grained MoE routing regimes, (ii) post-pretraining slimmability adaptation on top of existing pre-trained MoE checkpoints, and (iii) transfer beyond perplexity to zero-shot downstream reasoning benchmarks.

Figures 10 and 11 extend the main inference study to larger and finer-grained settings. On GPT2-Large (2.5B parameters, 50B tokens), MoSE continues to improve the inference-time Pareto frontier and achieves 32.8% lower FLOPs at matched perplexity. In finer-grained routing regimes, MoSE

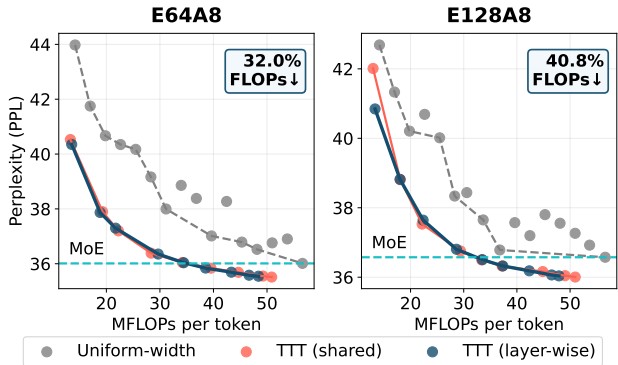

*Figure 11.* Pareto frontiers in finer-grained routing regimes. MoSE achieves 32.0% and 40.8% FLOPs reductions for E64A8 and E128A8, respectively, at comparable perplexity.

remains effective and yields 32.0% and 40.8% FLOPs reductions for E64A8 and E128A8, respectively.

To evaluate post-pretraining deployment, we additionally study continual-pretraining (CPT) slimmability adaptation starting from pretrained MoE checkpoints. Figure 12 shows that MoSE remains effective on DeepSeek-V2-Lite (16B), improving the inference-time Pareto frontier by 38.1% at comparable perplexity, corresponding to approximately 920 MFLOPs/token absolute savings. Detailed CPT adaptation results, including validation perplexity across widths and additional post-pretraining comparisons, are deferred to Appendix D.

Finally, Table 2 reports zero-shot downstream reasoning results on HellaSwag, PIQA, and SIQA. Across all three benchmarks, MoSE improves accuracy while reducing MFLOPs/token, indicating that the gains of width-adaptive execution are not limited to perplexity-based evaluation but transfer to downstream reasoning tasks as well. Additional runtime measurements and routing-stability diagnostics are provided in Appendix B.4 and Appendix B.5.

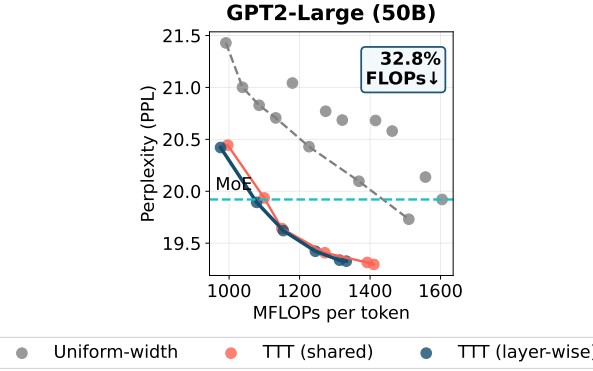

*Figure 10.* Pareto frontier on GPT2-Large (2.5B parameters, 50B tokens), where MoSE achieves 32.8% lower FLOPs at matched perplexity.

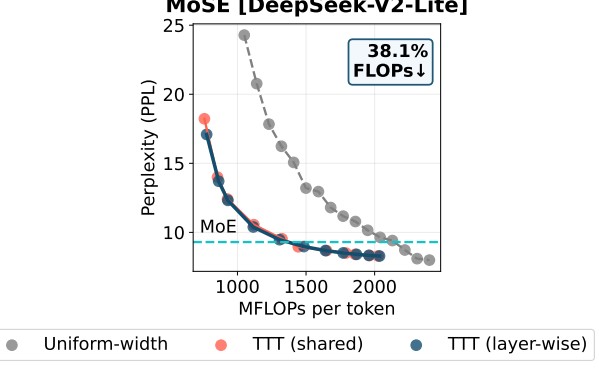

*Figure 12.* Pareto frontier on DeepSeek-V2-Lite (16B) after continual-pretraining slimmability adaptation. MoSE achieves 38.1% lower FLOPs at matched perplexity, corresponding to approximately 920 MFLOPs/token absolute savings.

*Table 2.* Zero-shot downstream reasoning results on the GPT2-Medium setting. MoSE improves accuracy while reducing inference compute on HellaSwag, PIQA, and SIQA.

| TASK | METHOD | ACCURACY | MFLOPs/TOKEN |
|---|---|---|---|
| HELLASWAG | MoE | 0.335 | 402.7 |
| HELLASWAG | MoSE (TTT) | **0.385** | **272.5** |
| PIQA | MoE | 0.675 | 402.7 |
| PIQA | MoSE (TTT) | **0.710** | **319.1** |
| SIQA | MoE | 0.290 | 402.7 |
| SIQA | MoSE (TTT) | **0.305** | **319.1** |

## 4. Related Work

**Mixture-of-Experts Transformers.** Mixture-of-Experts (MoE) models date back to classical conditional computation via expert specialization and learned gating (Jacobs et al., 1991; Jordan & Jacobs, 1994). Modern deep learning MoE revived this idea at scale by introducing sparsely-gated expert layers, enabling large parameter counts without proportional increases in per-token compute (Shazeer et al., 2017; Fedus et al., 2022). Early work on sparsely gated MoE layers demonstrated the feasibility of conditional computation, and subsequent works such as switch transformers (Fedus et al., 2022), GLaM (Du et al., 2022), and V-MoE (Riquelme et al., 2021) established MoE as a practical approach for large-scale transformer pretraining. Across these works, the dominant design choice is to replace transformer FFN layers with expert layers and select a fixed number of experts (e.g., top-$k$) per token. A common limitation of this paradigm is that once an expert is selected, it is executed at full capacity, even though the routing itself is sparse.

**Slimmable Networks.** Slimmable networks train a single set of parameters that can be executed at multiple widths, enabling runtime accuracy-efficiency trade-offs without training separate models (Rippel et al., 2014; Yu et al., 2019; Yu & Huang, 2019b; Horváth et al., 2021; Kusupati et al., 2022; Devvrit et al., 2023; Tastan et al., 2025). Related "train-once, deploy-many" approaches include once-for-all networks, which optimize a super-network that can be specialized into multiple subnetworks post hoc (Cai et al., 2020; 2025). These ideas have inspired a broad range of follow-up work, mostly focused on resource efficiency (Horváth et al., 2021; Mei et al., 2022), communication and computational efficiency (Wang et al., 2022), collaborative fairness (Tastan et al., 2025), and neural architecture search (Yu & Huang, 2019a).

**Elastic MoEs.** Recent work on elastic Mixture-of-Experts (MoE) tackles the long-standing rigidity of Top-k routing, where models trained with a fixed number of active experts fail to generalize when the inference budget changes. RoE (Roster of Experts) (Zibakhsh et al., 2025) approaches elasticity without retraining by injecting stochasticity into the router and aggregating multiple diverse forward passes, effectively turning a single MoE into a test-time ensemble; while this improves robustness and quality under larger compute budgets, it does not change the underlying expert representations and incurs additional inference costs proportional to the number of passes. In contrast, Elastic MoE (EMoE) (Gu et al., 2025) modifies training to expose experts to diverse co-activation patterns via stochastic sampling and hierarchical router regularization, enabling the model to tolerate larger inference-time k than seen during training; however, its gains depend on carefully designed routing losses and still assume a flat expert set without explicitly structuring expert importance.

MoSE bridges these lines of work by equipping MoE layers with slimmable widths, enabling conditional computation not only over which experts are activated but also over the internal capacity used by each activated expert.

## 5. Discussion & Limitations

MoSE provides an additional axis of conditional computation in MoE architectures, which allows for an additional degree of freedom in dedicating representational capacity at deployment time, paving the way for novel optimizations for LLMs. For example, MoSE could be leveraged as a mechanism of self-speculation (Leviathan et al., 2023; Miao et al., 2024; Cai et al., 2024; Tiwari et al., 2025). A lightweight "draft" model would be implicitly realized by executing the same model at reduced widths, thus accelerating inference without the need to retrain for explicit alignment. Additionally, in agentic settings (Wu et al., 2024; Tastan et al., 2026), an agent could select the model width depending on perceived task difficulty or uncertainty. Therefore, smaller widths could be used for easy steps, while additional computation could be leveraged under high-uncertainty and complex tasks. This equips MoE models with an adaptive dimension that can save on latency without sacrificing performance.

The current empirical validation covers GPT-style models trained from scratch, a larger 2.5B/50B setting, finer-grained routing regimes, zero-shot downstream reasoning benchmarks, and continual pre-training adaptation of DeepSeek-V2-Lite (16B). A remaining limitation is that we do not yet report full from-scratch training at frontier MoE scale.

## 6. Conclusion

In this paper, we have presented MoSE, a method that enables elasticity in MoE models by leveraging slimmable experts, allowing scaling of representational capacity and computation based on input needs and underlying hardware capabilities. We illustrate that MoSE is able to converge smoothly and improves the inference-time compute-quality frontier while preserving strong full-width performance.

## Impact Statement

This paper presents work aimed at advancing the field of Machine Learning. MoSE enables adaptive inference-time trade-offs between compute and model capacity in mixture-of-experts language models, which may improve accessibility and efficiency across deployment settings. At the same time, the slimmable nature of the model introduces an additional axis of conditional behavior, and future safety-alignment and robustness evaluations should account not only for the base model behavior but also for behavior across different execution widths and compute regimes. Beyond this consideration, we are not aware of any additional societal consequences that require specific discussion.

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

# A. Experimental Details

## A.1. Datasets and Evaluation Tasks

This section details the datasets used for pre-training and evaluation, the evaluation protocol, and the metrics we report.

**Pre-training.** All models are pre-trained on the OpenWebText corpus (Gokaslan et al., 2019), a large-scale corpus of English web text commonly used for training GPT-style language models. We use OpenWebText exclusively for pre-training in all experiments to ensure that differences in performance arise from model design and inference-time execution rather than changes in training data. We pre-train models for multiple token budgets depending on model scale and routing configuration (see Table 3). For continual pre-training (CPT) experiments, we initialize from a pretrained non-slimmable MoE checkpoint and continue on the same OpenWebText corpus with the same optimization and training recipe.

**Evaluation.** We evaluate both language modeling performance and zero-shot transfer on a set of standard benchmarks. All evaluations are conducted **zero-shot** (i.e., without any task-specific fine-tuning). We use the following datasets:

- **OpenWebText**: We report perplexity on a held-out split of OpenWebText to measure in-domain pre-training quality.

- **WikiText-103**: A Wikipedia-based language modeling benchmark used to assess out-of-domain generalization. We report perplexity on the standard evaluation split.

- **LAMBADA**: A dataset designed to test long-range context understanding via next-word prediction. We report both zero-shot **accuracy** (exact match next-word prediction) and **perplexity**. The validation split of this dataset consists of 5153 entries.

- **Winograd Schema Challenge (WSC)**: A pronoun resolution benchmark designed to test commonsense reasoning. We evaluate in the zero-shot setting and report accuracy. The validation split of this dataset consists of 273 entries.

- **HellaSwag:** A commonsense completion benchmark with four candidate endings. We evaluate in the zero-shot multiple-choice setting and report accuracy.

- **PIQA:** A physical commonsense reasoning benchmark with two candidate solutions. We evaluate in the zero-shot multiple-choice setting and report accuracy.

- **Social IQA (SIQA):** A social commonsense reasoning benchmark with three candidate answers. We evaluate in the zero-shot multiple-choice setting and report accuracy.

For all evaluations, we keep the pretrained model parameters fixed. We do not perform fine-tuning on any downstream task. For MoSE with test-time training, the only adapted parameters are the *sharpness* parameters $\gamma$ (either shared across transformer blocks or layer-wise), while all model weights, expert weights, and router parameters remain unchanged.

For MoSE with test-time training, we learn $\gamma$ by minimizing the language modeling loss over a short calibration stream $\mathcal{D}_{\text{calib}}$ (drawn from any available dataset, e.g., OpenWebText in our settings). The calibration optimizes only $\gamma$ under a given budget $\Gamma$, and then reuses the learned $\gamma^{\star}$ to evaluate the model under that budget. We emphasize that this procedure is lightweight because $\gamma$ is extremely low-dimensional (one scalar per layer in the layer-wise variant, or a single scalar in the shared variant). In our experiments, we limit the calibration dataset to only 50 batches, with a batch size of 6.

**Metrics.** We report the following metrics across datasets:

- **Perplexity (PPL):** For language modeling datasets (OpenWebText, WikiText-103, and LAMBADA), we report perplexity computed from the average negative log-likelihood. Lower is better.

- **Accuracy (Acc):** For LAMBADA and WSC, we report accuracy, corresponding to exact-match prediction.

**Compute reporting.** Alongside task metrics, we report inference-time compute in **MFLOPs per token**. Compute is measured by counting the floating-point operations incurred by the transformer forward pass, including expert computation under the given routing and width policy. We use MFLOPs per token to compare compute-quality trade-offs across inference modes and to trace Pareto frontiers by varying the budget $\Gamma$.

*Table 3.* Model architectures, routing settings, and training configurations used in the from-scratch full pre-training experiments.

| MODEL | ROUTING | PARAMS | LAYERS | HEADS | HIDDEN | TOKENS | BATCH | ITERS |
|---|---|---|---|---|---|---|---|---|
| GPT2-SMALL | E8A2 | 55M | 6 | 6 | 384 | 3B | 12×6×4 | 10K |
| | E8A2 | 55M | 6 | 6 | 384 | 15B | 12×6×4 | 50K |
| | E8A4 | 55M | 6 | 6 | 384 | 4.5B | 8×9×4 | 15K |
| | E16A4 | 83M | 6 | 6 | 384 | 7.5B | 8×9×4 | 25K |
| | E64A8 | 253M | 6 | 6 | 384 | 3B | 6×8×4 | 15K |
| | E128A8 | 480M | 6 | 6 | 384 | 3B | 6×8×4 | 15K |
| GPT2-STANDARD | E8A2 | 322M | 12 | 12 | 768 | 3B | 6×12×4 | 10K |
| | E8A2 | 322M | 12 | 12 | 768 | 15B | 6×12×4 | 50K |
| | E8A4 | 322M | 12 | 12 | 768 | 9B | 6×12×4 | 30K |
| GPT2-MEDIUM | E8A2 | 1B | 24 | 16 | 1024 | 7.5B | 2×12×4 | 75K |
| GPT2-LARGE | E8A2 | 2.5B | 36 | 20 | 1280 | 50B | 4×6×4 | 500K |

## A.2. Hyperparameters

This section describes the hyperparameters shared across all experiments. A summary of model architectures, routing settings, pretraining token budgets, and training schedules is provided in Table 3. Unless otherwise stated, the hyperparameters described below are held fixed across model sizes and routing configurations.

**Model architectures and routing.** We evaluate three GPT-style model scales: GPT2-Small, GPT2-Standard, GPT2-Medium, and GPT2-Large under multiple routing configurations, denoted by $EnAk$, where $n$ is the number of experts and $k$ is the number of activated experts per token. We additionally evaluate finer grained expert settings and continual pre-training adaptation on top of pretrained MoE checkpoints, including DeepSeek-V2-Lite (16B parameters). The specific combinations of model scale, routing setting, parameter count, and training token budget used in our experiments are summarized in Table 3. The sequence length is fixed to 1024 tokens for all experiments. These configurations are chosen to systematically vary routing capacity and model scale while maintaining comparable training settings.

**Pretraining optimization.** We optimize all models using the AdamW optimizer with a learning rate $6 \times 10^{-4}$, weight decay $10^{-1}$, and momentum parameters $(\beta_1, \beta_2) = (0.9, 0.95)$. Gradients are clipped to a maximum norm of 1.0. We use a linear learning rate schedule with warmup, followed by linear decay to a minimum learning rate of $6 \times 10^{-5}$. The warmup lasts for 200 iterations.

All runs use top-$k$ expert routing with auxiliary regularization; we apply a load-balancing loss with a coefficient 0.01 and a router $z$-loss with a coefficient 0.001. Training and evaluation capacities are set to 1.25 and 2.0, respectively, across runs with $k = 2$; when $k = 4$, we increase the capacities to 1.75 and 3.0 respectively.

**Slimmable training.** For MoSE, the hidden dimension of MoE experts is made slimmable. We set $w_{\min} = 0.25$ and $w_{\max} = 1.0$. For practical reasons, we discretize this range $\mathcal{A} \in [w_{\min}, w_{\max}]$ (see Equation 3) using a step size 0.05, resulting in $\mathcal{A} = \{0.25, 0.30, 0.35, \ldots, 1.00\}$. This strategy allows a single pretrained model to support a continuum of inference-time widths with minimal training overhead.

**Test-time training.** Test-time width identification is applied only after pre-training. The parameter $\gamma$ is optimized using a short calibration stream while keeping all pretrained model parameters fixed. The learned $\gamma$ is then reused for all subsequent inference-time evaluations. We fix the calibration set to OpenWebText everywhere, and we limit the calibration dataset to only 50 batches, with a batch size of 6. We use the SGD optimizer with a learning rate of 0.01 for this training.

**Compute.** The GPT2-Small, GPT2-Standard, and GPT2-Medium experiments reported in this paper were run using NVIDIA A100-SXM4-40GB GPUs. The larger GPT-Large and DeepSeek-V2-Lite continual pre-training experiments were run using NVIDIA RTX PRO 6000 Blackwell GPUs. Training is performed using DDP spanning 4 GPUs.

# B. Additional Analyses

## B.1. What do $\gamma$ values look like?

Figure 13 illustrates how the learned sharpness parameter $\gamma$ adapts to the computing budget (in terms of MoE FLOPs) and model scale. Across all models, $\gamma$ is optimized for the respective budget: low budgets favor larger $\gamma$ values, corresponding to more concentrated width allocation, while higher budgets drive $\gamma$ toward or below 1.0, yielding more uniform allocation across active experts.

When using a single shared $\gamma$, test-time training consistently finds a well-calibrated value for each budget that already delivers strong performance, aligning with the Pareto-optimal trends observed in earlier plots. The layer-wise variant is more expressive, assigning different sharpness levels to different transformer blocks, revealing non-uniform importance across layers, particularly at low budgets. This additional flexibility explains the improved compute-quality trade-offs observed with layer-wise width identification while maintaining stable behavior as the budget increases.

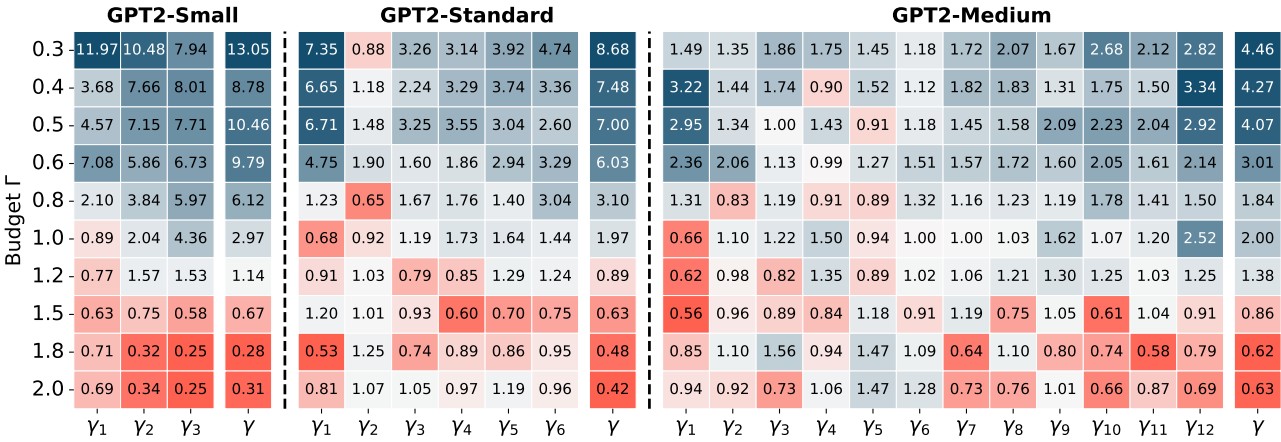

*Figure 13.* Learned $\gamma$ parameters across budgets and model scales under E8A2 setting. Visualization of the learned $\gamma$ parameters obtained via test-time training for MoSE under different routing budgets $\Gamma$. Each row corresponds to a budget and columns show either layer-wise $\gamma_\ell$ values (left of each pair) or a shared $\gamma$ (right of each pair).

## B.2. FLOPs Breakdown

Figure 14 illustrates the breakdown of inference-time floating-point operations (FLOPs) across model scales for GPT2-Small, GPT2-Standard, and GPT2-Medium. For each model, we decompose the total MFLOPs per token into contributions from MoE computation, attention, and other operations (linear layers, embeddings, etc.). The total MFLOPs per token are shown at the center of each donut chart.

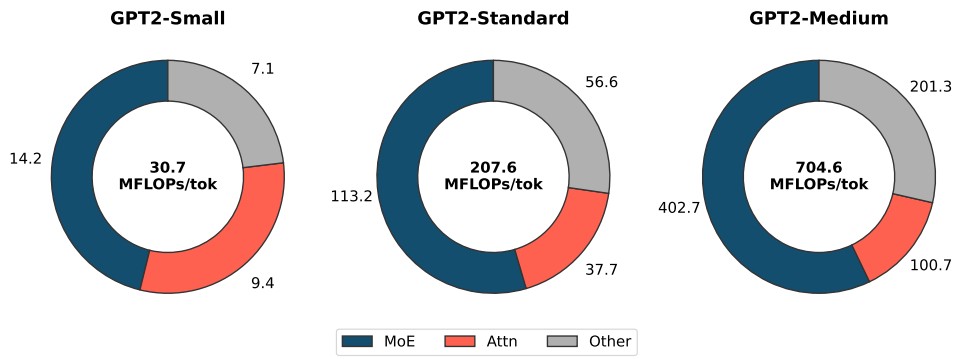

*Figure 14.* Inference-time FLOPs breakdown across model scales. Donut charts show the absolute MFLOPs per token attributed to MoE experts, attention, and other operations for GPT2-Small, GPT2-Standard, and GPT2-Medium. Total MFLOPs per token are shown at the center of each subplot.

Across all model sizes, MoE computation accounts for the largest fraction of the FLOPs, and this dominance becomes more pronounced as the model scale increases. While attention and other components grow with model size, their relative contribution remains substantially smaller than that of the MoE experts. For example, in GPT2-Small, expert computation already constitutes a significant portion of total compute, and by GPT2-Medium, MoE experts account for the majority of inference-time FLOPs.

This breakdown highlights why width adaptation is particularly effective for MoE models at scale. Since expert computation dominates the overall compute budget, reducing expert width directly targets the primary cost driver while leaving attention and routing mechanisms unchanged. As a result, MoSE is able to achieve substantial inference-time compute savings without modifying the underlying model architecture or routing behavior.

### B.3. Width-wise Pre-training Dynamics

Figure 15 demonstrates the validation perplexity dynamics of MoSE across different execution widths during pre-training. Across all model scales and training budgets, larger widths consistently converge faster and achieve lower perplexity, while smaller widths follow the same overall training trajectory with higher steady-state perplexity. Importantly, all widths exhibit stable and monotonic convergence, indicating that slimmable training does not introduce optimization instability even at reduced widths.

The relative ordering of widths is preserved throughout training: wider configurations dominate narrower ones at all stages, and the gap between widths narrows as training progresses. This behavior is consistent across both shorter (3B-token) and longer (15B-token) training regimes, as well as across model scales.

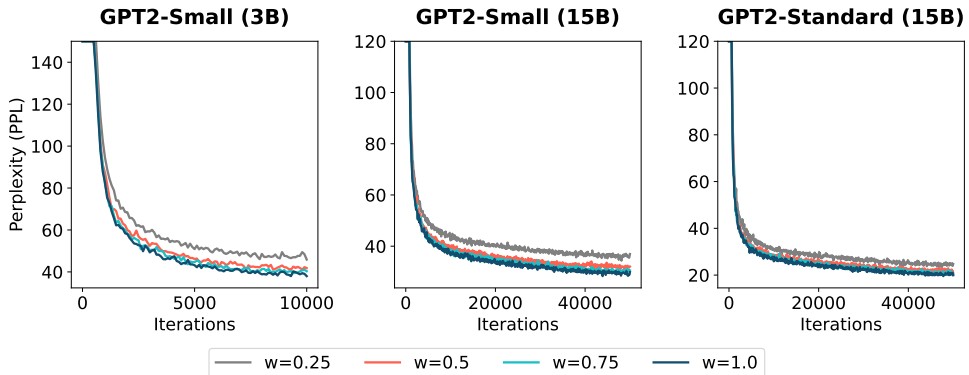

*Figure 15.* Validation perplexity dynamics across widths during pre-training. Validation perplexity as a function of training iterations for different execution widths $w \in \{0.25, 0.5, 0.75, 1.0\}$ during MoSE pre-training on OpenWebText. Results are shown for GPT2-Small trained for 3B (left) and 15B (middle) tokens, and GPT2-Standard trained with 15B tokens (right). For clarity, we clip early iteration values.

### B.4. Runtime Diagnostics

The main paper uses FLOPs/token as the primary hardware-agnostic measure of inference cost. This section justifies that choice empirically. In our implementation, latency and throughput track FLOPs closely across the evaluated operating points, so FLOPs is an informative proxy for end-to-end runtime in the regimes considered here.

These measurements are also important for interpreting the practicality of width adaptation. The inference-time width policy itself is lightweight: the sharpness parameter $\gamma$ is calibrated once, and the serving-time overhead only requires a simple transformation of the top-$k$ router probabilities before projection to executable widths. The runtime plots therefore test whether the algorithmic savings predicted by FLOPs are reflected in actual latency and throughput measurements.

We emphasize that these results justify the use of FLOPs for the current experimental setup; they do not claim that runtime must remain linear in FLOPs under all hardware or deployment regimes. In larger or more memory-bound systems, the realized wall-clock gains may differ, which is why we continue to present FLOPs as the primary architecture-level metric in the main body.

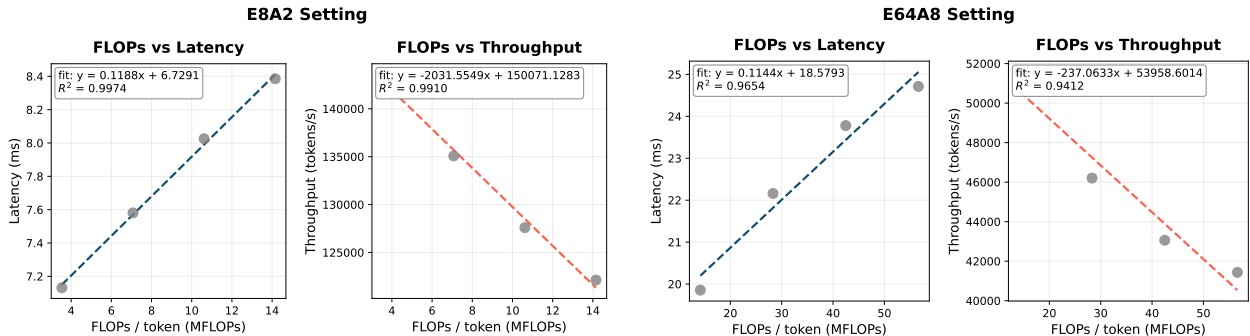

*Figure 16.* **Runtime analysis.** Latency and throughput track FLOPs closely in both E8A2 and E64A8 settings, supporting the use of FLOPs as the primary hardware-agnostic compute axis in the main paper.

### B.5. Routing Diagnostics

A natural concern with the MoSE training objective is whether exposing the same model to both full-width and sampled-width executions could distort router behavior, introduce expert imbalance, or destabilize optimization. Our expectation is that this should not happen because widths are *not* learned by the router during training: the router still learns standard expert selection under the usual MoE stabilization terms, while width allocation itself is only introduced at inference/calibration time.

The diagnostics in Figure 17 test this claim directly. We compare the full-width and sampled-width branches using three complementary statistics:

  (i)  the coefficient of variation (CV) of expert loads, which measures imbalance across experts;
 (ii)  normalized routing entropy, which measures whether routing collapses toward a small subset of experts; and
(iii)  the auxiliary/load-balancing loss, which directly tracks the regularization used to avoid expert collapse.

Similar trajectories across the two branches indicate that the multi-width objective does not induce a new routing pathology.

We additionally visualize expert-utilization heatmaps (Figure 18) and compare global expert-selection histograms across widths (Figure 19). These analyses are meant to address a stronger question than simple training stability: not only does MoSE train stably, but the sampled-width branch preserves nearly the same routing structure as the full-width branch, supporting the interpretation that Equation 6 adds width robustness without perturbing the learned routing policy.

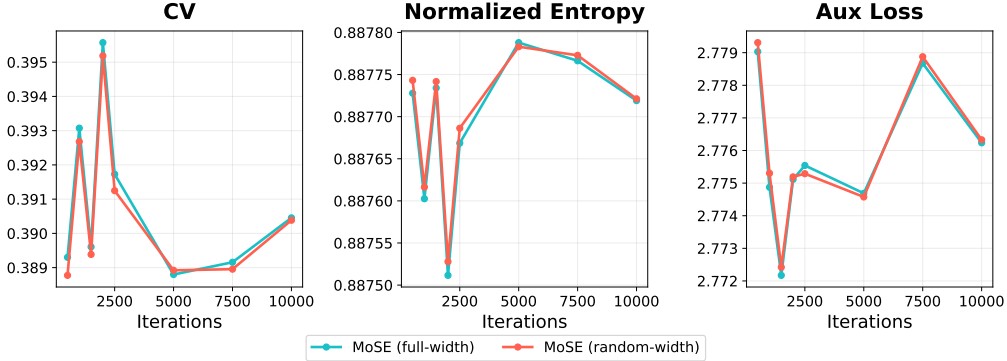

*Figure 17.* **Routing-stability diagnostics** for the full-width and sampled-width branches during training. We report coefficient of variation (CV) of expert loads, normalized routing entropy, and auxiliary/load-balancing loss. The close match between the two branches indicates that the multi-width objective does not induce routing instability.

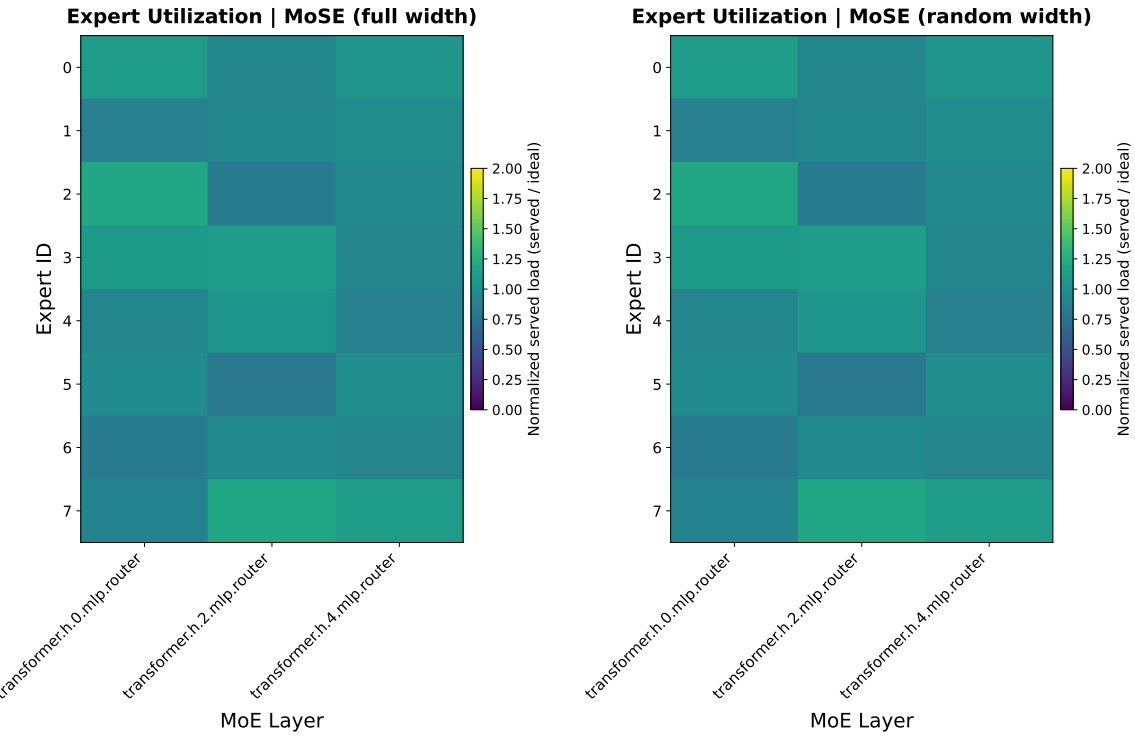

*Figure 18.* Expert-utilization heatmaps for the full-width and sampled-width branches. Expert usage is nearly identical across branches, with no dead experts or severe overload, indicating that width sampling does not alter the qualitative routing pattern learned by the model.

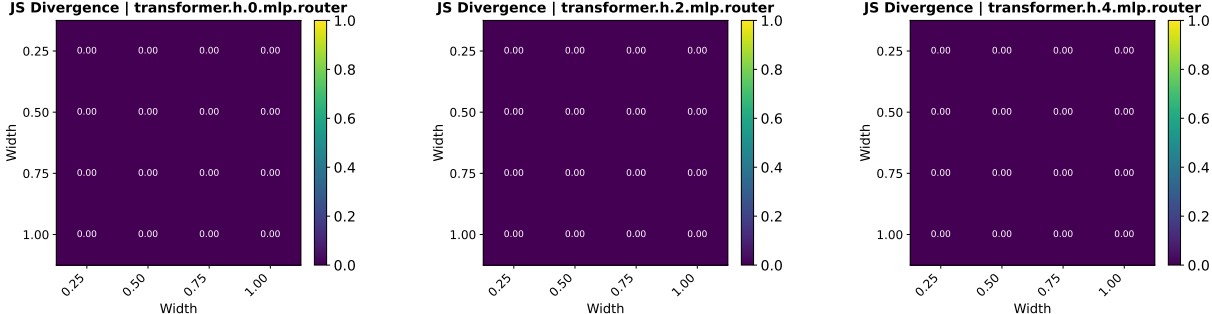

*Figure 19.* JS divergence between global expert-selection histograms across width pairs. We aggregate routed expert assignments over a fixed validation shard and compare the resulting histograms across widths. Near-zero divergence indicates that global routing distributions remain unchanged across widths.

## C. Additional Results

### C.1. Increased Expert Count and Pre-Training Tokens

The main-body inference results already show that MoSE improves the compute-quality frontier across model scales and routing settings. This section isolates a complementary regime in which both the active expert count and the pre-training token budget are increased.

Specifically, Figure 20 focuses on E8A4 setting and compares the GPT2-Small and GPT2-Standard models trained with an increased amount of pretraining data (4.5B and 9B pretraining tokens, respectively). Under this setting, MoSE with test-time training continues to outperform across both model sizes. This confirms our findings from Figures 4, 5, and 6 and supports the robustness of MoSE across routing settings, model sizes, and data scales.

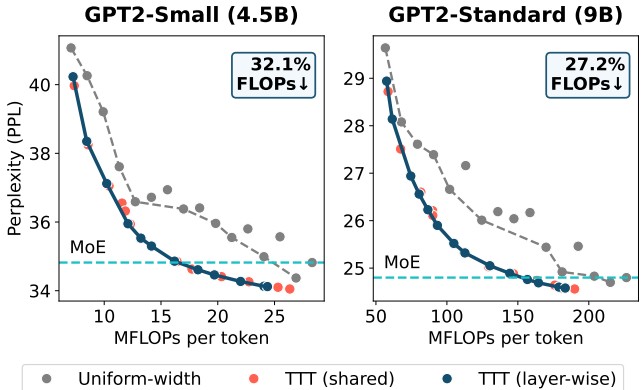

*Figure 20.* MoSE under E8A4 setting and increased pre-training tokens. We compare inference-time execution modes for the GPT2-Small (4.5B tokens) and GPT2-Standard (9B tokens) models. MoSE (TTT) maintains a clear advantage over the uniform-width mode across both model sizes.

## D. Continual Pre-Training Results

This section provides additional results for continual pre-training (CPT) slimmability adaptation, where we start from a pretrained MoE checkpoint and continue training under the MoSE objective. These experiments complement the main-body DeepSeek result in Figure 12 and isolate the practical regime where retraining from scratch is too expensive, but a pretrained MoE checkpoint already exists.

We view full multi-width pre-training and CPT as complementary regimes rather than substitutes. Full pre-training yields

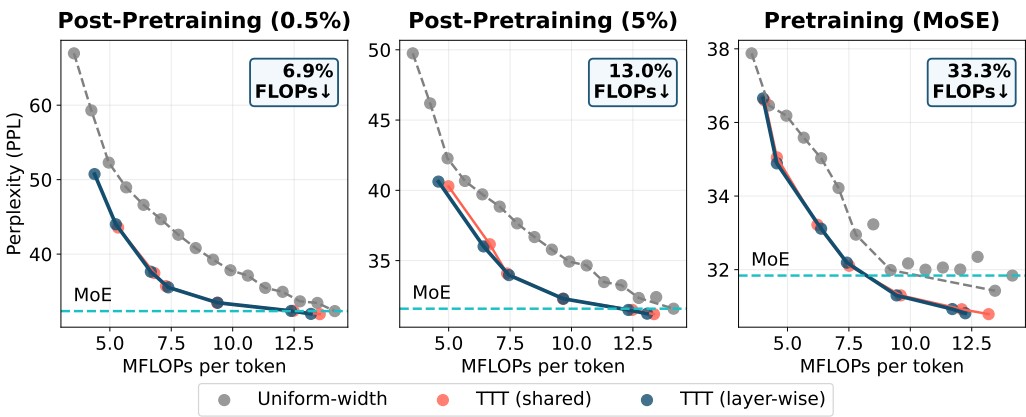

*Figure 21.* Post-pretraining slimmability adaptation compared with full MoSE pretraining. Left: 0.5% continued pretraining. Middle: 5% continued pretraining. Right: full MoSE pretraining. Even short CPT phases recover meaningful inference-time savings, while full pretraining yields the strongest compute-quality frontier.

*Figure 22.* Validation perplexity across widths during continual pre-training adaptation. The full-width model remains stable, while reduced-width subnetworks improve rapidly and become usable early in adaptation.

the strongest compute-quality frontier because width robustness is learned from the start. CPT, by contrast, asks how much of that benefit can be recovered post hoc with a short continued-training phase, and whether slim-width subnetworks can be enabled without degrading the original full-width checkpoint.

Figure 21 addresses the first question. Even a $0.5\%$ continued pre-training budget already recovers measurable inference-time savings, while $5\%$ continued pre-training yields a $13.0\%$ FLOPs reduction and substantially narrows the gap to full MoSE pre-training. Figure 22 addresses the second question: the full-width validation curve remains stable throughout adaptation, while slimmed-width subnetworks improve rapidly and become usable early in training. Together, these results show that slimmability can be induced post hoc under the same training recipe, providing a lightweight deployment path when full pretraining is impractical.

