# OpenReview forum: "MoSE: Mixture of Slimmable Experts for Efficient and Adaptive Language Models"
_ICML.cc/2026/Conference — ICML 2026 regular_

### Official Review · Reviewer_uQCG · 2026-03-08

**Soundness:** 2
**Presentation:** 3
**Significance:** 2
**Originality:** 3
**Overall Recommendation:** 3
**Confidence:** 4

**Summary:**

This paper proposes MoSE, a special MoE architecture where each expert is designed with a slimmable structure that can operate at variable widths. The authors introduce a training recipe combining multi-width training with standard MoE objectives and explore several inference strategies for determining expert widths, including a lightweight test-time training approach based on router confidence. Experiments on GPT-style models trained on OpenWebText show that MoSE can match or outperform standard MoE models while achieving better accuracy–compute trade-offs.

**Compliance With Llm Reviewing Policy:**

Affirmed.

**Final Justification:**

The rebuttal addressed parts of my concerns but the concern of model size still remains. I raise my score from 2 to 3.

**Key Questions For Authors:**

## Questions
1. What advantages does the slimmable width design bring when combined with MoE models compared to dense models?
2. What is the effect of loss design in pretrain stage on stability?
3. What is the function of *per-token width budget* in Section 2.4?

## Minor typos

1. In the definition of $S(x)$ (Section 2.4, line 120), it may need to be $G(x)_i \neq 0$.

**Limitations:**

yes

**Strengths And Weaknesses:**

## Strengths
1. The paper is easy to follow.
2. The figures in paper are clear and help illustrate the proposed method.
3. The paper conducts comprehensive experiments and ablation study.


## Weaknesses

1. The novelty of the work appears limited and the motivation is unclear. The connection between the *slimmable width* and MoE models is not sufficiently well justified. Why is it applied on MoE rather than dense model?
2. Lack of deep analysis of the loss design (Equation 6) in pretrain stage, which may lead to unstable optimization.
3. Lack of detailed explaination of some design in MoSE Inference (Section 2.4), e.g., the function of *per-token width budget*, the motivation of the design Equation 7-9.
4. The advantage over baselines is relatively modest, which somewhat limits the significance of the proposed method.
5. In the experiments, this work focuses on relatively small models (less than 1B parameters), while MoE architectures are typically applied at much larger scales.

---

> ### Author Rebuttal · Authors · 2026-03-31
>
> Thank you for the detailed review and for pinpointing the places where the original submission needed clearer explanation.
>
> > W1
> > Q1. slimmable width design
>
> MoE and slimmability address two different decisions. In standard MoE, routing decides which experts are relevant, but once selected, each expert is still executed at fixed internal capacity. Slimmability adds a second axis by allowing the model to control how much capacity each selected expert receives.
>
> This is particularly effective in MoE is that expert outputs are already weighted by routing probabilities. Lower-probability experts contribute less to the final output and can therefore be evaluated at reduced capacity with limited effect on performance. MoSE explicitly aligns **expert importance (routing probability)** with **compute allocation (width)**, which does not arise naturally in dense transformers. In addition, MoSE directly targets the main compute bottleneck of MoE models: the hidden dimension expansion (4d) (see Appendix B.2).
>
> > W2
> > Q2. loss design
>
> The original paper already provides initial evidence that Eq. (6) is stable: Figure 3 shows that MoSE tracks MoE closely during training, and full-width MoSE remains competitive with standard MoE. Eq. (6) uses one full-width pass and one random-width pass: the former preserves standard MoE behavior, while the latter trains the same experts to remain useful under reduced capacity.
>
> Importantly, the router is **not** learning a token-level width policy during training; it still learns standard expert selection under the same MoE stabilization terms, while width allocation is learned only at test/calibration time. Also, the two passes do not mix capacities; they are standalone passes.
>
> To address this concern more directly, we now add additional routing diagnostics for both the full-width and random-width branches: CV of expert loads, normalized routing entropy, and aux/LBL loss (see https://ibb.co/BVfjC0b7). These closely match each other throughout training. We also include expert-utilization heatmaps for both branches (see https://ibb.co/WNZMzknr), which show nearly identical usage patterns and no dead or overloaded experts. Together, these results indicate that Eq. (6) does not introduce instability. Please also see Figures 3 and 13.
>
> > W3
> > Q3. per-token width budget in Section 2.4
>
> The per-token width budget $\Gamma$ specifies how much total expert capacity can be spent on a token, measured in units of full-expert width. Eqs (7)-(9) then determine how that total budget is distributed across the selected experts. Specifically, Eq. (7) normalizes router probabilities over the selected set, Eq. (8) applies the sharpness parameter $\gamma$, and Eq. (9) renormalizes the result into allocation weights. A smaller $\gamma$ makes the allocation closer to uniform; a larger $\gamma$ concentrates more width on higher probability experts. Eq. (10) then scales by $\Gamma$, and Eq. (11) projects the result to valid widths. In short, $\Gamma$ is an input deciding how much compute is available, and $\gamma$ controls how that compute is distributed. We will rewrite Section 2.4 to make this intuition explicit.
>
>
> > W4. The advantage over baselines is relatively modest, which somewhat limits the significance of the proposed method.
>
> We would like to clarify our claim: we are not focused on performance/full-width quality jump, but a better inference-time Pareto frontier from a single model. To strengthen this point, we now add three new results.
>
> 1. On GPT2-Large / 2.5B parameters / 50B tokens, MoSE achieves 32.8% FLOPs reduction to achieve the same perplexity as MoE. That is: an absolute reduction of 525 MFLOPs per token. (see https://ibb.co/XkzyvkmX)
>
> 2. In more fine-grained expert settings, MoSE achieves 32% and 40.8% FLOPs reduction for E64A8 and E128A8, respectively. (see https://ibb.co/7JXFkZ3f)
>
> 3. We add a post-pretraining slimmability adaptation experiment (see https://ibb.co/VYfRhgPT, https://ibb.co/gFhtc8mP), where we start from a pretrained MoE checkpoint and continue under the MoSE objective; with only 5% continued pretraining, this already yields 13% FLOP reduction while preserving the full-width model. We believe these additions materially strengthen the significance argument.
>
> > W5. In the experiments, this work focuses on relatively small models (less than 1B parameters), while MoE architectures are typically applied at much larger scales.
>
> We now address this directly by expanding both the model scale and the expert-count scale. Specifically, we add a 2.5B-parameter / 50B-token result and two higher-expert-count settings, E64A8 and E128A8.
>
> Thank you for your detailed review. We hope these responses clarify the necessary points and provide sufficient clarity to our method. If so, we would greatly appreciate a revised rating.

---

> > ### Author Rebuttal · Reviewer_uQCG · 2026-04-02
> >
> > Thanks for the author's detailed response and most of my concerns are addressed clearly.
> >
> > However, the concern about model size still remains. The authors provide the results of a 2.5B-parameter model which is still relatively small. In practice, MoE models are typically deployed at much larger size. I understand that there may be computational resource constraints. Could authors provide the results of the model with just more than 10 B params?
> >
> > Anyway, I am willing to **raise my score from 2 to 3**.

---

> > > ### Author Response · Authors · 2026-04-03
> > >
> > > Thank you for the follow-up comment. We appreciate that most of your concerns were addressed clearly, and we also thank you for already updating your score. We especially appreciate your continued willingness to further reconsider the assessment.
> > >
> > > To address this directly, we evaluated MoSE on **DeepSeek V2 Lite (16B parameters)** by applying continual pretraining slimmability adaptation. This provides a substantially larger and more modern MoE test case than the GPT-style models.
> > >
> > > The new result is encouraging: MoSE still improves the inference-time Pareto frontier, achieving **38.1% FLOPs reduction**, corresponding to an **absolute savings of about 920 MFLOPs/token**. We attach the Pareto plot for this setting here (https://ibb.co/Tx8wXBCC).
> > >
> > > We also include the per-width validation perplexity curves during continual pretraining on DeepSeek V2 Lite (see https://ibb.co/hFk8CZZJ). These show that the full-width model remains stable, while the slim-width subnetworks become usable quickly, consistent with our earlier findings.
> > >
> > > We believe this additional result strengthens the remaining scaling argument and that MoSE is not limited to sub-10B models, but also remains effective on a modern DeepSeek MoE model. We thank you again for your thoughtful consideration. If this additional evidence resolves the remaining concern about model size, we would be very grateful if you would consider further updating your score.

---

### Official Review · Reviewer_su8P · 2026-03-08

**Soundness:** 3
**Presentation:** 3
**Significance:** 2
**Originality:** 3
**Overall Recommendation:** 5
**Confidence:** 4

**Summary:**

This paper proposes Mixture of Slimmable Experts (MoSE), an extension of Mixture-of-Experts (MoE) architectures that enables conditional computation not only over which experts are selected but also over how much capacity each activated expert uses. In standard MoE models, routing sparsely selects experts, but each selected expert is executed at full capacity, resulting in coarse-grained accuracy–compute trade-offs. MoSE addresses this limitation by equipping each expert with a nested slimmable structure whose internal width can be dynamically adjusted, allowing the model to control the amount of computation allocated within each activated expert. The authors introduce a training procedure that supports multi-width execution under sparse routing and propose inference-time width allocation strategies, including router-conditioned allocation and a lightweight test-time training method that learns how to map routing signals to expert widths under a fixed compute budget. Experiments on GPT-style language models trained on OpenWebText show that MoSE matches or improves the performance of standard MoE at full width and achieves better accuracy–compute trade-offs by shifting the Pareto frontier toward lower inference FLOPs.

**Compliance With Llm Reviewing Policy:**

Affirmed.

**Final Justification:**

The twice rebuttals answer my main concerns and it deserved a 5 score for me.

**Key Questions For Authors:**

1.How sensitive is the width selection policy to hyperparameters such as routing thresholds or compute budgets?

2.Would reinforcement learning or optimization-based routing policies further improve the adaptive width selection? If you have tried other non-heuristic method to determine width, show the results are good for improving the contribution. If not, it's also OK, but pls tell me why the method you use is good enough.

3.How does MoSE scale to very large MoE models (e.g., hundreds of experts or multi-billion parameter models)?

4.Could the slimmable design introduce load-balancing issues among experts during training?

**Limitations:**

No. The paper briefly acknowledges that the experiments are limited to small-scale LLMs and that it remains unclear whether similar multi-width behavior can be induced at post-training time, but the discussion of limitations is still quite narrow. It would be helpful for the authors to more explicitly discuss potential failure modes of router-conditioned width allocation, the uncertainty around scaling to modern large MoE models, and the additional implementation complexity that variable-width experts may introduce at inference time.

I fully understand that due to experimental limitations, this method has only been validated on a small scale. However, even when its shortcomings are acknowledged, this still diminishes the paper's contribution. Nevertheless, I remain willing to maintain a positive evaluation at this initial stage.

**Strengths And Weaknesses:**

I have summarized the strengths and weaknesses here. For convenience, authors need not address each shortcoming listed individually during the rebuttal period, simply respond to my questions directly.

Soundness. The method is technically reasonable and the overall training recipe is coherent. The experiments support the claim that MoSE improves the perplexity and FLOPs trade-off across several model sizes and routing settings. However, the key design choices, especially the probability-based width allocation and the test-time training mechanism, are mostly justified empirically rather than analytically.

Presentation. The paper is generally clear and easy to follow. The motivation, architecture, and training procedure are described in a structured way, and the figures help illustrate the main idea. That said, the inference-time width allocation strategy could be explained more clearly, especially the role of the sharpness parameter \gamma and the intuition behind the allocation formula.

Significance. The paper addresses a relevant limitation of standard MoE models, namely that selected experts are always executed at full capacity. Introducing expert width as an additional axis of conditional computation is practically meaningful and could be useful in deployment settings with varying compute budgets. Still, the demonstrated gains are moderate and the experiments are limited to relatively small-scale models, which makes the broader impact somewhat uncertain.

Originality. The main originality comes from combining MoE routing with slimmable experts so that computation can vary both across experts and within each expert. This is a sensible and potentially useful extension of existing ideas. However, the contribution is best viewed as an incremental architectural combination rather than a fundamentally new modeling paradigm.

---

> ### Author Rebuttal · Authors · 2026-03-31
>
> Thank you for the careful and balanced review. We especially appreciate your clear framing of the paper and your guidance to respond directly to the main questions.
>
> Before answering the questions, we will clarify Section 2.4 in the revision as follows:
> $\Gamma$ is the **total per-token width budget**, measured in units of full-expert widths, i.e., it controls **how much total compute** is allocated across selected experts. $\gamma$ controls **how concentrated** that budget is: smaller $\gamma$ gives a flatter allocation across selected experts, while larger $\gamma$ concentrates more width on higher-probability experts. Eqs (7)-(9) convert router scores into a normalized allocation, Eq. (10) scales that allocation to the total budget $\Gamma$, and Eq. (11) clips/discretizes it into legal execution widths.
>
> TL;DR: $\Gamma$ decides how much compute is available; $\gamma$ decides how it is distributed.
>
> > Q1. How sensitive is the width selection policy to hyperparameters such as routing thresholds or compute budgets?
>
> Our method does not introduce an additional routing threshold; it uses the same top-k routing mechanism as the MoE baseline. If the reviewer refers to k, this is already examined in the paper (see Fig. 8.)
> The primary new control variable is the width budget $\Gamma$, together with the concentration parameter $\gamma$. In practice, the Pareto frontiers are obtained by sweeping the compute budgets, and we observe stable improvements across multiple routing settings and model sizes; we now further strengthen this with larger scale results.
> In the learned TTT setting, $\gamma$ is not manually tuned; it is automatically calibrated on a short calibration set under a fixed budget. This reduces deployment-time sensitivity, adapting directly to the target compute constraint.
>
> > Q2. Would reinforcement learning or optimization-based routing policies improve ...
>
> Our TTT method is already an **optimization-based** width-allocation mechanism, albeit intentionally low-dimensional: it learns $\gamma$ (shared or layerwise) by minimizing LM loss under a fixed budget while keeping all model parameters frozen. We chose this design because it isolates the value of width adaptation, keeps calibration cheap, and avoids conflating the contribution with a separate high-capacity policy model. We also tried a larger controller (e.g., an MLP); but it overfits to the calibration set more easily and did not preserve rank-stability of expert preferences as robustly as the $\gamma$-based approach.
> The current ablation already shows a clear hierarchy: uniform width < fixed normalized-probability allocation < learned TTT. This suggests that the important ingredient is learning the probability-to-width mapping under a budget, rather than adding a more complex controller. We agree that richer controllers or RL-style policies are interesting future work, but they are beyond the scope of our work, which takes the first step towards slimmable MoE models. We thank the reviewer for this valuable point and for suggesting a new direction.
>
>
> > Q3. How does MoSE scale to very large MoE models (e.g., hundreds of experts or multi-billion parameter models)?
>
> We now address both axes directly. On the model-size axis, we add a GPT2-Large / 2.5B-parameter / 50B-token experiment (see https://ibb.co/XkzyvkmX), where MoSE achieves 32.8% FLOPs reduction (**525 MFLOPs/token absolute reduction**). On the expert-granularity axis, we add E64A8 and E128A8 routing settings (see https://ibb.co/7JXFkZ3f), with 32% and 40.8% reduction, respectively. These additions materially strengthen the scaling story beyond the original experiments.
>
>
> > Q4. Could the slimmable design introduce load-balancing issues among experts during training?
>
> We do not observe such a pathology empirically. During training, width is not learned or predicted by the router; the router still learns standard expert selection, and we do not mix width branches; they have separate load/capacity regimes and activate the same experts and exhibit the same behaviors. To test this, we report three diagnostics for the full-width and sampled width branches (see https://ibb.co/BVfjC0b7). All of them closely match each other. We also include expert-utilization heatmaps (see https://ibb.co/WNZMzknr), which show that expert usage is nearly identical across the full-width and sampled-width settings, with no dead experts or severe overload. Together, these results support the claim that slimmability does not introduce a new load-balancing issue during training.
>
> We also include JS divergence plots between expert-selection histograms across widths (https://ibb.co/j9BDfzTZ), where we aggregate routed expert assignments over the fixed validation shard and compare the resulting histograms between width pairs. As shown in the figure, all entries are exactly zero, indicating that the global routing distribution remains unchanged across widths.
>
> Thank you again for your thoughtful feedback.

---

> > ### Author Rebuttal · Reviewer_su8P · 2026-04-01
> >
> > Most of my concerns have been solved, and the only thing is that the 2.5b scale model is also a small one, and it's also gpt-series model, so i am curious about if this method could be applyied in more new and larger model, such as qwen3-30b-a3b moe, NVIDIA Nemotron-3-Nano-30B-A3B, or DeepSeek-V2-Lite, i will consider raise my score from 4 to 5 if you can try one of them.

---

> > > ### Author Response · Authors · 2026-04-03
> > >
> > > Thank you for the follow-up comment. We appreciate your suggestion to test MoSE on a more recent and larger MoE model.
> > >
> > > In response, we evaluated MoSE on **DeepSeek V2 Lite** by applying continual pre-training (CPT) slimmability adaptation, where we use a pretrained non-slimmable MoE model as an initialization. This directly addresses the question of whether the method remains effective beyond the GPT-style models in the main submission.
> > >
> > > As shown in the attached experiments, MoSE still improves the inference-time frontier, achieving **a 38.1% reduction in FLOPs**, corresponding to an **absolute reduction of about 920 MFLOPs/token**. We include the Pareto plot here (https://ibb.co/Tx8wXBCC). This suggests that the main benefit of MoSE is not tied to the GPT-series architecture, but extends to a newer large-scale MoE model as well.
> > >
> > > We also include the per-width validation perplexity curves (see https://ibb.co/hFk8CZZJ). It shows that the full-width MoE model remains stable, and the slim-width subnetworks become usable quickly, consistent with our earlier findings.
> > >
> > > We thank the reviewer again for this suggestion and for considering updating their score.

---

### Official Review · Reviewer_6enM · 2026-03-10

**Soundness:** 3
**Presentation:** 3
**Significance:** 3
**Originality:** 3
**Overall Recommendation:** 4
**Confidence:** 4

**Summary:**

This paper proposes MoSE (Mixture of Slimmable Experts), an extension of MoE for language models. By adding slimmable experts, the model can adjust both expert selection and computation width, enabling a finer accuracy–compute trade-off. Experiments show improved or comparable performance to standard MoE at the same FLOPs.

**Compliance With Llm Reviewing Policy:**

Affirmed.

**Final Justification:**

Thank the authors for their detailed responses to my concerns in the rebuttal. I retain my score and tend to acceptance.

**Key Questions For Authors:**

- Does the router implicitly learn to favor certain widths or experts during training?

- How does MoSE perform when applied to larger-scale MoE models?

- Does dynamic width selection introduce additional latency or complexity in real-world inference pipelines?

**Limitations:**

See the weaknesses.

**Strengths And Weaknesses:**

Pros:

1. MoSE introduces slimmable experts, allowing each expert to run at different widths while keeping the original routing mechanism largely unchanged, making it easy to integrate into existing MoE frameworks.

2. The model can adjust both the number of active experts and their widths at inference time, while multi-width training ensures stable performance across configurations.

3. Experiments show a better accuracy–FLOPs trade-off than standard MoE under similar compute budgets.

Cons:

1. The method mainly combines MoE and slimmable networks, with little explanation of why this combination works particularly well.

2. The runtime width decision relies on simple heuristics rather than more principled approaches like learned controllers or budget-aware routing.

3. Results are shown on moderate-size models, so it is unclear how well the method scales to very large MoE systems.

4. The paper focuses on FLOPs, but gives little discussion of practical factors like latency, GPU utilization, or runtime overhead.

---

> ### Author Rebuttal · Authors · 2026-03-31
>
> Thank you for the positive and constructive review. We especially appreciate that you highlighted the simplicity of integration and the improved quality-FLOPs tradeoff, while also pushing us on deployment realism and larger-scale relevance.
>
> > W1. The method mainly combines MoE and ...
>
> MoSE introduces a second form of conditional computation that is complementary to MoE routing. Standard MoE uses top-k gating to decide **which experts** to activate, but once selected, each expert is executed at fixed capacity. MoSE adds control over **how much capacity** each selected expert receives.
>
> This is  particularly natural in MoE because expert outputs are already weighted by routing probabilities. Lower-probability experts contribute less to the final output and can therefore be executed at reduced capacity with limited effect on performance. MoSE thus aligns **expert importance (routing probability)** with **compute allocation (width)**, which does not arise in dense transformers. As a result, slimmability is especially well matched to MoE and directly improves the compute/quality trade-off.
>
> > W2. simple heuristics
>
> We would like to clarify that the paper includes both a heuristic baseline and a learned, budget-aware controller. The normalized-probability rule is the fixed heuristic baseline, whereas TTT is a learned controller that optimizes the sharpness parameter $\gamma$ under a fixed-width budget while keeping model weights frozen. We intentionally keep this controller low-dimensional to make the adaptation lightweight and interpretable. In deployment, it is learned once and reused; serving-time overhead is limited to computing $p_i^{\gamma}$ over the top-k router probs, followed by normalization and projection.
>
> Empirically, we observe a clear progression: **uniform allocation < probability-based allocation < learned TTT**, indicating that even this simple controller captures most of the available gains. Richer controllers are interesting future work, but the current design is a minimal and practical instantiation of the core idea.
>
> > W3. moderate-size models
>
> We agree that the current experiments do not fully establish behavior at frontier-scale MoE systems.
>
> Our primary goal is not to target the largest possible MoEs, but to enable continuously tunable inference budgets from a single model, which we view as especially relevant for small-to-mid-scale language models (SLMs) deployed under **constrained or dynamic compute environments** (e.g., on-device or latency-sensitive serving). We will revise the paper to make this scope explicit.
>
> Within this intended scope, we nevertheless extend the original experiments along two axes:
>
> - a GPT2-Large / 2.5B-parameter / 50B-token experiment (see https://ibb.co/XkzyvkmX), where MoSE achieves 32.8% FLOPs reduction (525 MFLOPs/token absolute reduction).
> - more routing settings with E64A8 and E128A8 experts (see https://ibb.co/7JXFkZ3f), where the gains remain strong at 32% and 40.8%, respectively.
>
> Across these settings, MoSE consistently improves the compute-quality frontier, suggesting favorable empirical scaling behavior with both model size and expert-count granularity.
>
> > W4. The paper focuses on FLOPs, discussion on latency, GPU utilization, or runtime overhead.
>
> In our runtime analyses, latency and throughput track FLOPs linearly (see https://ibb.co/8DMRwPmw). This is why we used FLOPs as the main hardware-agnostic axis in the paper: in our setup, it is a strong proxy for end-to-end runtime. We will include these results in the revision together with GPU utilization measurements (approx. 53%/57.5%/61%/64% for the respective FLOPs).
>
> That being said, we recognize that this relationship may change in larger or more memory-bound settings. In such cases, we expect our width adaptation to become more appealing, as it will further ease the memory traffic bottleneck. We aim to include this discussion in the manuscript.
>
>
> > Q1. Does the router implicitly learn to favor certain widths or experts during training?
>
> During pretraining, the router is **not** learning a token-level width policy. The multi-width objective exposes the model to two capacity regimes (a full-width branch and a sampled-width branch), but the router still learns standard expert selection under the standard MoE stabilization terms. Width allocation itself is learned at test-time/calibration time through $\gamma$.
>
> To verify that this does not create a routing pathology, we now include (i) expert-utilization heatmaps for the full-width and sampled-widths (see https://ibb.co/WNZMzknr), which show nearly identical expert-usage patterns and no dead or overloaded experts, and (ii) training diagnostics for CV, normalized entropy, and aux loss (see https://ibb.co/BVfjC0b7), which closely match between the two branches throughout training.
>
> Thank you again for the encouraging review. We believe the added responses make MoSE much stronger.

---

> > ### Author Rebuttal · Reviewer_6enM · 2026-04-03
> >
> > Thank the authors for their detailed responses to my concerns in the rebuttal.

---

> > > ### Author Response · Authors · 2026-04-06
> > >
> > > Thank you very much for the follow-up and for indicating that your concerns have been fully resolved. We greatly appreciate your careful reading of the rebuttal and your positive assessment of our responses.
> > >
> > > If you feel that the clarifications and additional experiments now support a stronger overall assessment than your original score reflects, we would be very grateful if you could update it accordingly.
> > >
> > > Thank you again for your time and thoughtful evaluation.

---

### Official Review · Reviewer_FMGr · 2026-03-12

**Soundness:** 2
**Presentation:** 3
**Significance:** 2
**Originality:** 3
**Overall Recommendation:** 4
**Confidence:** 4

**Summary:**

This paper takes the standard MoE setup and makes each expert slimmable. You can run any activated expert at a fraction of its full width by slicing the intermediate FFN dimension. During training, for each batch they do two forward passes and average the losses. At inference, they explore three modes for setting expert widths (uniform width, width proportional to router probabilities, and a lightweight TTT). They train GPT-2 scale models (55M to 1B) on OpenWebText with various configurations and show that MoSE matches standard MoE at full width.

**Compliance With Llm Reviewing Policy:**

Affirmed.

**Final Justification:**

The authors resolved most of my concerns with additional experimental results and thoughtful responses. I was especially not sure about the justifiabliity of the overhead, but given the proposed method's benefits (increased accuracy and efficiency), I will increase the score.

**Key Questions For Authors:**

Regarding W3, what do you think about using your approach to finetune the already pretrained LLMs? I'm not sure if frontier labs will spend more costs with two forward passes just to pretrain a massive MoE with continuously tunable inference budget. However, people still might be interested in finetuning the pretrained LLMs to support that perhaps? That way, you can experiment this method on more interesting tasks and models.

Overall, the concern rises because it's not often justifiable to spend more 1.5~2x costs in the *pretraining* regime. In a finetuning regime, perhaps. May I ask why you chose to tackle the pretraining problem. Are there any relevant papers that already tackled the finetuning regime? Also, can you provide more analysis on how much computational overhead your method incurs?

**Limitations:**

yes.

**Strengths And Weaknesses:**

**Strengths**
- I think it's a fairly simple method for a clear problem
- The main advantage is that they allow you to really flexibly determine the budgets you want to spend per inference.
- The experiments show that it can match or improve the full-width MoE with lower costs.
- They include some robustness-style analyses like scaling training tokens, changing routing settings, and transferring learned ε across
    datasets.

**Weaknesses**

W1. The experiments mostly show the inference-time computational savings. But the pretraining step of this method requires two forward passes and how much computational overhead it incurs is not clear.
W2. The experiment scope is limited. They are only conducted on old GPT2 variants and few metrics like perplexity and accuracy. It's not clear how they can scale to downstream metrics and long horizon, agentic tasks.
W3. Recent MoEs from DeepSeek, for example, are more fine-grained than ever. In the past, MoEs used to choose 2 out of 8 experts. Now, they choose 8 out of 64, 128, etc. experts, which are fine-grained enough. In this new regime where strong models are pretrained with *already slim* experts, it's not clear how this paper can be meaningfully significant and interesting.
W4. Minor writing issue: pg 2. inside 2.4 (a sparse set S(x) (where Gxi = 0))

---

> ### Author Rebuttal · Authors · 2026-03-31
>
> Thank you for the thoughtful review. We especially appreciate the perspective on whether MoSE can be applied more economically.
>
> > W1. inference-time savings, pretraining forward passes; computational overhead?
>
> We thank the reviewer for raising this important point. In the full pretraining setting, the overhead ranges from 1.5x to 2x per update, since each step includes one full-width pass and one sampled-width (slim) pass. While this increases training cost, it yields a single model that supports a wide range of inference budgets without retraining.
>
> When full pretraining is not feasible, MoSE also supports a lightweight **continual-pretraining (CPT) adaptation** regime, discussed under W3 (which can be performed in ~10 minutes on a single GPU). We view this CPT regime as particularly relevant for small- and mid-scale language models deployed under dynamic or constrained compute budgets. And, we view full-pretraining and CPT as complementary regimes. We will revise the paper accordingly.
>
> > W2. The experiment scope .. downstream metrics and long horizon, agentic tasks.
>
> We agree that the original scope was limited, and we will explain the claim accordingly. The paper's main claim is about **language-model quality vs. inference compute**, not long-horizon or agentic-task performance, which we  noted as future work. To strengthen the scope along this axis, we now add a **GPT2-Large/ 2.5B-parameter model pretrained on 50B tokens** (see https://ibb.co/XkzyvkmX), where MoSE continues to improve the Pareto frontier and achieves a 32.8% FLOPs reduction (**525 MFLOPs absolute reduction**) at the same perplexity. We also broaden the routing regime to high-expert-count settings, discussed below.
>
> To address downstream relevance more directly, we also include zero-shot reasoning benchmarks (HellaSwag, PIQA, and SIQA) using the GPT2-Medium model:
>
> | Task       | Method   | Accuracy   | MFLOPs/token |
> |-----|-------|-------|------|
> | HellaSwag  | MoE  | 0.335 | 402.7 |
> | | MoSE (TTT)  | **0.385**  | **272.5** |
> | PIQA | MoE  | 0.675 | 402.7 |
> | | MoSE (TTT) | **0.710**  | **319.1** |
> | SIQA | MoE | 0.290 | 402.7 |
> | | MoSE (TTT) | **0.305**  | **319.1** |
>
> Across all tasks, MoSE improves accuracy while reducing inference compute, showing that the benefit extends beyond perplexity to downstream reasoning benchmarks.
>
> > W3. Recent MoEs ... 8 out of 64, 128 ...
>
> > Also, why study pretraining rather than adapting already pretrained LLMs?
>
> We address both points directly below.
>
> First, we respectfully clarify that 'many experts' (fine-grained) should not be conflated with 'slim/slimmable experts'. Even in fine-grained MoEs such as E64A8/E128A8, each selected expert still uses the standard hidden dimension expansion (4d) and is executed at full internal capacity once selected. MoSE is therefore orthogonal to expert count: routing decides which experts are active, while MoSE additionally determines how much of each selected expert to use. To show this directly, we now include E64A8 and E128A8 settings (see https://ibb.co/7JXFkZ3f); MoSE still improves the frontier, with 32% and 40.8% reduction at comparable perplexity.
>
> Second, MoSE supports two complementary usage scenarios:
> (i) full pretraining with multi-width objectives, which yields the strongest compute-quality trade-offs, and
> (ii) continual-pretraining slimmability adaptation, where an existing MoE checkpoint is augmented with width elasticity at significantly lower cost.
>
> We focus on the pretraining in the main paper because it establishes the method in a general setting where slimmability is integrated into the learned representations. However, when compute is scarce and a pretrained MoE already exists, the second regime is more practical.
>
> To evaluate this, we added a **continual-pretraining slimmability adaptation** experiment: continued pretraining from a pretrained MoE checkpoint under the MoSE objective with a single randomly sampled width. The new results show that sub-width experts become usable quickly while the full-width model remains stable, and with 5% continued pretraining (CPT), the adapted checkpoint already achieves a 13% FLOPs reduction (this attached plot uses only about 10 minutes and a single A100 GPU). See https://ibb.co/VYfRhgPT, https://ibb.co/gFhtc8mP.
>
> This shows that slimmability can be enabled post hoc, without retraining from scratch. We therefore view the two regimes as complementary: full pretraining provides maximal gains, while continual pretraining offers a lightweight practical path.
>
> We thank the reviewer again for raising practical questions about training overhead, fine-grained MoE relevance, and the applicability of continual pretraining. We believe the new 2.5B/50B, E64A8/E128A8, and continual-pretraining adaptation results directly address the key open questions. If the reviewer finds these additions helpful, we would be grateful if they would consider updating their assessment accordingly.

---

> > ### Author Rebuttal · Reviewer_FMGr · 2026-04-03
> >
> > Dear authors, thank you for providing your thoughtful responses and providing additional experiment results. Your comments mostly resolved my concerns. The major remaining concern is still whether the extra overhead for pretraining cost is justifiable for the purpose of having "a wide range of inference budgets." Is this something really desired for the overhead? However, given that the proposed method provides a higher accuracy and better efficiency, I will increase a score accordingly.

---

> > > ### Author Response · Authors · 2026-04-03
> > >
> > > Thank you for the follow-up comment, and we appreciate that our responses and additional experiments resolved most of your concerns.
> > >
> > > We agree that the key remaining question is when the additional training overhead is justified in practice. Our view is that there are now two distinct deployment regimes.
> > >
> > > First, full multi-width pretraining is the right regime when one wants the strongest possible compute quality frontier from a single model. In this setting, the extra cost is the one-time overhead we reported earlier, with the second pass being a slim-width pass rather than a second full-width pass. The alternative is to train models with smaller experts separately, which would incur a higher cost than full training, several times over for each model, compared to roughly 1.75x total overhead for training an elastic family of MoEs.
> > >
> > > Second, when that overhead is difficult to justify, MoSE now also supports continual pretraining (CPT) adaptation on top of an existing MoE model. This provides a substantially cheaper path to enable slimmability, while still yielding meaningful inference-time savings. We therefore do not view the method as requiring full retraining in all practical settings.
> > >
> > > The new experiment we conducted with the DeepSeek V2 Lite model (**16B** parameters) (see https://ibb.co/Tx8wXBCC, https://ibb.co/hFk8CZZJ) makes this trade-off clearer. In this model, the MoE layers account for roughly **2399 / 2779 MFLOPs/token (~86%)** of total inference cost, so reducing expert computation directly addresses the dominant bottleneck. After training with MoSE, it achieves **38.1% FLOPs reduction**, corresponding to an **absolute savings of about 920 MFLOPs/token**. In this regime, the benefit is not merely "supporting a wide range of budgets" in the abstract; it is a substantial reduction in the dominant inference cost of a large modern MoE.
> > >
> > > This is also why we believe the training overhead question should be considered together with the target deployment regime. For smaller or medium-scale deployments, CPT offers a lightweight path; for larger MoEs where expert computation dominates inference cost, the absolute savings become much more compelling.
> > >
> > > We thank the reviewer again for pushing on this practical point. We believe the added continual pretraining (an excellent suggestion from the reviewer) and DeepSeek V2 Lite results materially strengthen the answer to the question of when the overhead is justified, and we would appreciate it if the reviewer could consider updating their score.

---

### Decision · Program_Chairs · 2026-04-30

**Decision:**

Accept (regular)

**Comment:**

This paper proposes MoSE, a mixture of slimmable experts framework for improving efficiency and adaptability in large language models. Reviewers generally agreed that the idea is well motivated, technically sound, and clearly presented. The paper reports consistent improvements in efficiency while maintaining competitive performance across a range of evaluation settings.

The overall consensus among reviewers is positive. The rebuttal helped clarify several implementation details and strengthened the empirical evaluation. Reviewers did not identify any major technical flaws, and most concerns raised in the initial reviews were adequately addressed during discussion.

The experimental results provide sufficient support for the main claims, and the proposed approach appears relevant to practical settings where efficiency and adaptability are important.

Overall, the paper makes a useful contribution. Therefore, I recommend accepting the paper.